# Dwell Times, Wavepacket Dynamics, and Quantum Trajectories for Particles with Spin 1/2

**DOI:** 10.3390/e26040336

**Published:** 2024-04-14

**Authors:** Bill Poirier, Richard Lombardini

**Affiliations:** 1Department of Chemistry and Biochemistry and Department of Physics, Texas Tech University, Lubbock, TX 79409, USA; 2Department of Physics, St. Mary’s University, San Antonio, TX 78228, USA

**Keywords:** dwell times, wavepacket dynamics, quantum trajectories, de Broglie–Bohm theory, foundations of physics, quantum control, entanglement

## Abstract

The theoretical connections between quantum trajectories and quantum dwell times, previously explored in the context of 1D time-independent stationary scattering applications, are here generalized for multidimensional time-dependent wavepacket applications for particles with spin 1/2. In addition to dwell times, trajectory-based dwell time *distributions* are also developed, and compared with previous distributions based on the dwell time operator and the flux–flux correlation function. Dwell time distributions are of interest, in part because they may be of experimental relevance. In addition to standard unipolar quantum trajectories, bipolar quantum trajectories are also considered, and found to relate more directly to the dwell time (and other quantum time) quantities of greatest relevance for scattering applications. Detailed calculations are performed for a benchmark 3D spin-1/2 particle application, considered previously in the context of computing quantum arrival times.

## 1. Introduction

This article represents the coalescing of several different idea threads. First and foremost, there is the idea of *quantum time*, which has long been controversial—as evidenced, e.g., by the plethora of different quantum time quantities available on the market [1,2,3,4,5,6,7,8,9,10,11,12,13,14]. It is the general considered opinion that quantum mechanics does not endow time with a Hermitian operator—at least not like other quantum observables [8,15,16,17]. So how does one define, say, the collision time between two quantum particles? What does that quantity mean precisely, and can it be experimentally measured? Then there is the idea of *quantum trajectories*, going back to Madelung and Bohm [18,19,20,21,22,23,24,25]. Given that there are trajectory-based ways to formulate or interpret quantum mechanics—where definite positions and momenta of all particles can be determined over time—this seems a natural avenue towards understanding the nature of quantum time. And yet, one of the most reliable quantum time metrics currently in use—called the “dwell time” [2,7,26]—is generally inconsistent with the corresponding quantum trajectory traversal time, even for the most straightforward case of one-dimensional (1D) time-independent scattering. Although recent efforts demonstrate that it is, indeed, possible to compute dwell times (and other related quantum time quantities) using quantum trajectories [5,6], these works also hint at the notion that an alternate trajectory-based approach, using what are called *bipolar* quantum trajectories [27,28,29,30,31,32,33], may be more appropriate. If so, would this imply that quantum particles actually follow bipolar, rather than conventional “unipolar”, quantum trajectories [34]?

In this work, we explore all of the above questions, not only in the context of 1D time-independent scattering with specific left- (or right-) incident boundary conditions, but also more generally, for many-dimensional and/or time-dependent wavepacket applications, with arbitrary initial conditions. Furthermore, we consider the interesting case of *spin-1/2 particles*. The issue of quantum time in the context of spin-1/2 particle dynamics is a very topical—and controversial—one that has been vigorously debated in a very recent set of publications [35,36,37,38]. At issue, it appears, is, once again, the question of whether a given time quantity (in this case, the arrival time) can in principle be measured experimentally or not. This is of interest, in part because quantum trajectory arrival time statistics show starkly different features than would otherwise be predicted by the quantum flux of the Pauli equation—and hence could help validate experimentally the “reality” of quantum trajectories.

We wish to be clear at the outset: the present work *in no way addresses or resolves the above controversy*. We focus solely on the (Pauli-equation-based) quantum dynamics itself, and take a wholly agnostic view with regards to the issue of subsequent experimental measurement—apart from offering a mild admonishment against saying “never”. Too many “no-go” theorems from mathematics have been undone in practice by clever workarounds. Indeed, even within the context of quantum time, we find one such very cogent example, in the form of Pauli’s famous theorem that would seem to argue against the existence of self-adjoint time operators conjugate to the Hamiltonian operator [39]. If not “undone”, per se, this theorem has been severely undermined in work by Galapon [40], which, moreover, has led to a better understanding of the theory of canonical commutation relations.

Be that as it may, we do adopt a stance insofar as the choice of quantum time quantity is concerned. As mentioned, the earlier work with spin-1/2 particles has concentrated on computing the distribution of particle *arrival times*, rather than *dwell times*. The former has been described as an “instant” quantity, with the latter characterized as an “interval” quantity [7]. Perhaps, more precisely, the arrival time is measured from an initial *time* to a final *place*. At least in the references above, an ensemble of quantum trajectories is distributed across a range of positions, (x,y,z), at initial time t=0, and allowed to propagate over time. The *arrival time* for a given trajectory is then defined as the time at which said trajectory crosses the z=L plane—with the trajectory ensemble as a whole thus providing a distribution across many such arrival time values. In contrast, the *dwell time* describes how much time the particle spends within the interval, $\left[z_{L},z_{R}\right]$. In many circumstances, this amounts to how much time it takes for the particle to *traverse* the interval, i.e., as measured from an initial *place* (or surface), $z=z_{L}$, to a final *place*, $z=z_{R}$ (or vice versa). That said, the dwell time can also incorporate “reflections”, for which the particle exits the interval, turns around, and passes through the interval again, traveling in the other direction.

The dwell time quantity offers a number of practical benefits. For example, the boundary conditions are consistent, in the sense that dwell time represents a “place-to-place” rather than “time-to-place” transition. Also, to the extent that one *can* construct a corresponding time operator, the dwell time operator has nice properties (e.g., it is Hermitian, and commutes with the Hamiltonian) [7]. In the time-independent scattering context, the dwell time is also very closely related to other quantum time quantities, such as the time delay matrix and the Smith lifetime matrix, and also the scattering *S* matrix [2,5,6,7]. Moreover, the dwell time can also be easily generalized for time-dependent wavepacket dynamics as well.

Most intriguingly, for our purposes anyway, is the close connection between the dwell time and the flux–flux correlation function (FFCF) discovered by Pollak and Miller [41]. The latter is based on the quantum mechanical flux operator, which, in turn, is motivated by *classical trajectories* (despite being a fully quantum entity). Importantly, it is not the dwell time, per se, that enters in here, but, rather, the *average* dwell time (which is also what relates most closely to the diagonal Smith lifetime matrix elements [2,5,6]). As we shall see, the dwell time itself often oscillates with respect to the interval endpoints, due to quantum interference between incident and reflected waves. By averaging the dwell time over these interference oscillations, cleaner asymptotic behavior can be achieved, as has been recognized for many decades.

In any event, the above facts—coupled with the fact that dwell times often do not agree with their corresponding unipolar quantum trajectory traversal times—have motivated the present dwell time formulation in terms of *bipolar* quantum trajectories [27,28,29,30,31,32,33]. The term “bipolar” refers to a wavefunction decomposition,
(1)$$\psi =\psi_{+}+\psi_{-},$$
where $\psi_{+}$ and $\psi_{-}$ are traveling waves headed in opposite directions. Whereas ψ itself may show significant interference, the bipolar $\psi_{\pm }$ components generally do not. The corresponding bipolar quantum trajectories, derived separately from $\psi_{\pm }$ rather than from ψ, are accordingly smooth and well behaved. In particular, the fact that bipolar quantum trajectories are (generally) nonoscillatory suggests that these might be better suited to obtaining (average) dwell times than are unipolar quantum trajectories. More compellingly, however, bipolar quantum trajectories are *classical-like*, and approach their true classical trajectory counterparts in the classical limit of large action. In any event, the fact that FFCF theory is also classical-like, thus, strongly suggests (at least to us) a close connection between bipolar quantum trajectories and quantum dwell times.

This connection is explored and developed in the present work, and then investigated in the specific context of the benchmark spin-1/2 three-dimensional (3D) wavepacket system proposed by Das and Dürr [36]. In addition to the aforementioned arrival time measurement controversy, this system is of interest for the different trajectory dynamics that ensue, depending on whether the initial spin state is a ${\hat{\sigma }}_{z}$ or a ${\hat{\sigma }}_{x}$ eigenstate. The quantum trajectories are different, despite the fact that the the time-evolving probability density functions are the same in both cases (because the Hamiltonian itself has no explicit spin dependence). For this system, we first solve for the unipolar quantum trajectories, using a specialized ensemble [33,42,43,44,45,46,47,48] from which arrival time distributions may be easily obtained.

After confirming agreement with the previous study [36], we then compute dwell time distributions using the unipolar quantum trajectories. Additionally, we compute *bipolar* wave components, $\psi_{\pm }$, which are then used to compute bipolar quantum trajectories, from which accurate dwell times for the $\psi_{\pm }$ component waves are also obtained. Finally, symmetry, together with an alternate interpretation in terms of wave reflection at z=0, are used to obtain a bipolar-trajectory-based (average) dwell time distribution for ψ itself.

## 2. Theory

### 2.1. Dwell Times

#### 2.1.1. Time-Independent

The dwell time quantity will first be defined for time-independent stationary scattering solutions of the one-dimensional (1D) Schrödinger equation,
(2)$$\left[-\frac{\hbar^{2}}{2m}\frac{d^{2}}{dx^{2}}+V\left(x\right)\;\right]\phi_{k}\left(x\right)=E\;\phi_{k}\left(x\right).$$
For convenience, we focus on 1D systems, although the generalization to the many-dimensional case is straightforward. Equation (Equation 2) above encompasses both left- and right-incident scattering solutions, respectively, through the positive and negative values of the wavenumber parameter, $k=\pm \sqrt{2mE}/\hbar$. The potential energy function, V(x), is thus presumed to vanish in both asymptotes, x→±∞.

The “dwell time” $\tau_{k}$, for a given time-independent scattering solution $\phi_{k}\left(x\right)$, is defined as the integrated probability density over the region of interest $\left[x_{L},x_{R}\right]$, divided by the incident flux:(3)$$\tau_{k}=\frac{1}{|j_{\mathrm{in}}|}\int_{x_{L}}^{x_{R}}\phi_{k}^{*}\left(x\right)\phi_{k}\left(x\right)\;dx$$
Note that in the definition of $\tau_{k}$, the directionality matters, with left-incident flux (i.e., $j_{\mathrm{in}}>0$) presumed for k>0, and right-incident flux presumed for k<0 (for a discussion of the limit as k→0, consult Ref. [7]). This is fitting and appropriate for left- and right-incident scattering solutions, as presumed above.

What happens, however, for time-independent superposition states, i.e.,
(4)$$\psi_{k}\left(x\right)=A_{+}\phi_{k}\left(x\right)+A_{-}\phi_{-k}\left(x\right),\;(\mathrm{for}\;k>0)\;?$$
Presumably, Equation (Equation 3) still holds. Alternatively, we might imagine additive contributions, i.e., $|A_{+}{|}^{2}\tau_{k}+{\left|A_{-}\right|}^{2}\tau_{-k}$, with suitable normalization in effect (i.e., $\langle \phi_{k}|\phi_{k}^{\prime }\rangle =\delta \left(k-k^{\prime }\right)$). We will return to this question in Section 3.1, as it impinges on the use of bipolar trajectories.

Note the generality of the dwell time quantity, in that it can be defined over *any* desired interval, $\left[x_{L},x_{R}\right]$. In practice—and especially with regard to matching with other quantum time and scattering matrix quantities—the interval is often chosen to include the entire scattering potential, extending sufficiently far into each asymptotic region. Under these circumstances, we have
(5)$$\tau_{k}=\frac{P_{T}\left(k\right)}{\left|j\right|}\int_{x_{L}}^{x_{R}}\phi_{k}^{*}\left(x\right)\phi_{k}\left(x\right)\;dx,$$
where *j* is the overall flux (which, like $j_{\mathrm{in}}$, is constant over all *x*), and $P_{T}\left(k\right)=j/j_{\mathrm{in}}$ is the transmission probability for the solution $\phi_{k}$. Note that $P_{T}\left(k\right)={\left|T\left(E\right)\right|}^{2}$, where T(E) is the (energy-dependent) transmission amplitude, although the latter quantity itself is not used or computed here.

Another important aspect about the time-independent dwell time that ought to be considered, especially when the interval extends across the entire scattering range, is the fact that $\tau_{k}$ oscillates with the interval limits, $x_{L}$ (e.g., for $\phi_{k>0}$) or $x_{R}$—since the density itself oscillates in one asymptote or the other. For this reason, it can be useful to define an *average* dwell time quantity $\langle \tau_{k}\rangle$, where the effective average is taken in some suitable manner (e.g., over one complete asymptotic cycle), such that changes in $\tau_{k}$ become proportional to changes in both $x_{R}$ and $x_{L}$. Indeed, it is $\langle \tau_{k}\rangle$ rather than $\tau_{k}$, per se, that is most directly related to the other quantum time quantities—especially the diagonal Smith lifetime matrix elements, here labeled $Q_{kk}$ (with k=±|k|), which we have previously argued are the most fundamental quantum time quantities [6]. In any event, one very straightforward way to define such a $\langle \tau_{k}\rangle$ quantity is through the use of bipolar quantum trajectories, and so we defer any further discussion to Section 2.2.4 and Section 3.2.1.

As a final clarifying comment, we point out that what we refer to above as the “average dwell time” $\langle \tau_{k}\rangle$ is by no means the same quantity which some other authors (notably Muga and coworkers [7]) call by the same name. Their usage of this phrase refers to what is here denoted $\tau_{k}$—which is regarded to be an “average” (or “mean”, in our parlance) because it can be obtained as the first moment over a suitable dwell time *distribution*. Particularly given that the concept of dwell time distributions has been proposed to be amenable (possibly) to experimental measurement—and given that the distribution from the standard definition is different from that which arises naturally in the quantum trajectory picture—further discussion of dwell time distributions is provided in Section 2.3.

#### 2.1.2. Time-Dependent

Let the time-dependent wavepacket, ψ(x,t), be a solution of the time-dependent Schrödinger equation,
(6)$$\left(-\frac{\hbar^{2}}{2m}\frac{d^{2}}{dx^{2}}+V\left(x\right)\right)\psi \left(x,t\right)=i\hbar \frac{d}{dt}\psi \left(x,t\right),$$
that is normalized to unity (i.e., ${\int \;|\psi \left(x,t\right)|}^{2}\;dx=1$ for all *t*). Then, the dwell time τ for ψ(x,t) over the interval $\left[x_{L},x_{R}\right]$ is defined via the following simple and intuitive expression [2,7,49,50]:(7)$$\tau =\int_{-\infty }^{\infty }\left(\int_{x_{L}}^{x_{R}}{\left|\psi \left(x,t\right)\right|}^{2}dx\right)dt$$

Given the normalization condition, Equation (Equation 7) above clearly represents the amount of time that the particle spends within the interval, as discussed. Incidentally, both Equation (Equation 7) and the earlier Equation (Equation 3) can be derived from the same Hermitian dwell time *operator*, $\hat{T}$:(8)$$\hat{T}=\int_{-\infty }^{\infty }\exp\left(i\hat{H}t\right)\;\left[\int_{x_{L}}^{x_{R}}|x\rangle \langle x|dx\;\right]\;\exp\left(-i\hat{H}t\right)\;dt,$$
where $\hat{H}$ is the Hamiltonian operator. Interestingly, $\hat{T}$ commutes with $\hat{H}$, rather than satisfying the expected canonical commutation relation. In any event, given their common origin in $\hat{T}$, it is perhaps not surprising that the time-independent $\tau_{k}$ and time-dependent τ quantities can be related to each other, provided that ψ(x,t) satisfy some additional properties [7].

In particular, we generally require that only left-incident stationary solutions, $\phi_{k>0}\left(x\right)$, contribute to the spectral decomposition of ψ(x,t), as follows:(9)$$\psi \left(x,t\right)=\int_{0}^{\infty }\;\langle \phi_{k}|\psi_{0}\rangle \exp\left(-i\hbar k^{2}t/2m\right)\phi_{k}\left(x\right)\;dk$$
In Equation (Equation 9) above, $\langle x|\psi_{0}\rangle =\psi \left(x,t=0\right)$ is the initial wavepacket, presumed to lie (with any significant probability) to the left of the scattering center. Note that exactly these additional, left-incident wavepacket conditions, as described above, are also presumed in the theoretical development of the bipolar wavepacket decomposition, $\psi =\psi_{+}+\psi_{-}$, as discussed in Ref. [31].

Assuming that the above left-incident wavepacket conditions also hold, we may also write [7,49,50]
(10)$$\tau =\int_{0}^{\infty }\;{\left|\langle \phi_{k}|\psi_{0}\rangle \right|}^{2}\tau_{k}\;dk.$$
This is an eminently reasonable relationship, which simply states that the dwell time for the left-incident wavepacket ψ(x,t) is the probabilistically weighted sum of the time-independent dwell times $\tau_{k}$ for the corresponding left-incident spectral states $\phi_{k}\left(x\right)$ that make it up. Additionally, since the initial wavepacket $\psi_{0}\left(x\right)$ is situated in the left asymptote where V(x)=0, the spectral decomposition $\langle \phi_{k}|\psi_{0}\rangle$ is simply a Fourier transform. Furthermore, since only k>0 states contribute, $|\psi_{0}\rangle =|\psi_{0+}\rangle$—i.e., the $\psi_{-}\left(x,t\right)$ wave only comes into existence at later times, as a result of $\psi_{+}\left(x,t\right)$ encountering the scattering potential [31] (note that later, when discussing free-particle wavepackets, t=0 will correspond to $\psi_{0}\left(x\right)$ in the vicinity of x=0).

Like its time-independent analog $\tau_{k}$, the time-dependent dwell time τ can be defined over any desired interval, $\left[x_{L},x_{R}\right]$. *Unlike* $\tau_{k}$, however, τ does *not* tend to oscillate with asymptotic variation of $x_{L}$ or $x_{R}$. At least if the above left-incident wavepacket conditions are employed, the reflected wave $\psi_{-}\left(x,t\right)$ only comes into being and propagates into the left asymptotic region at late times *t*—i.e., long after the incident $\psi_{+}\left(x,t\right)$ wave has already passed through this region. The two waves are thus *time-separated*, and therefore do not interfere with one another in this region. This represents a *substantial* difference from the stationary state case—suggesting that, asymptotically at least, τ and its oscillation-averaged cousin $\langle \tau \rangle$ (introduced in the time-independent context in Section 2.1.1) should agree for suitable wavepackets. In any event, this difference will prove to have major ramifications for the corresponding quantum trajectories.

### 2.2. Quantum Trajectories

#### 2.2.1. Introduction

The quantum trajectory picture attempts to frame quantum mechanics in a context that strongly resembles classical mechanics. In particular, the time evolution of quantum systems is described by trajectories, for which all particles have definite positions *and* momenta at all times. The Heisenberg uncertainty principle is not, however, violated, either because the quantum trajectory for a given system is guided in part by the wavefunction, or because the wavefunction itself is replaced with an *ensemble* of interacting trajectories [33,42,43,44,45,46,47,48], depending on the particular approach (with the latter, incidentally, being first introduced in a festschrift in honor of E. Pollak [42]).

Although quantum trajectories satisfy their own Newton-like evolution laws, there are two “short cuts” that may be profitably used for obtaining them, when the solution wavefunction ψ(x,t) is known *a priori*, as is the case here. The first approach is to compute the velocity field v(x,t) directly from the wavefunction via [21,22]
(11)v(x,t)=ℏmIm∂ψ(x,t)∂x/ψ(x,t),
and then integrate over time to obtain quantum trajectories. This results in one trajectory traveling through every spacetime point. As a consequence, these “unipolar” trajectories do not cross.

Although written in 1D form, Equation (Equation 11) above can be generalized in straightforward fashion to arbitrary system dimensionality, using gradients. In contrast, the second “short cut” applies only to 1D time-dependent wavepacket applications. Here, the well-known property that probability is conserved along quantum trajectories [20,21] (which also holds true for spin-1/2 systems) implies that a given quantum trajectory x(t) must satisfy [42,44]
(12)$$C=\int_{0}^{x}\;{\left|\psi \left(x^{\prime },t\right)\right|}^{2}\;dx^{\prime }$$
for some constant *C*.

In this manner, a given trajectory x(t) or entire trajectory ensemble, x(C,t), may be easily and directly obtained from the time-evolving probability density ${\left|\psi \left(x,t\right)\right|}^{2}$. Note that *C* for a given trajectory is constant over time; therefore, this parameter serves as a useful trajectory labeling coordinate. In practice, when performing numerical calculations, it is convenient to discretize the x(C,t) ensemble uniformly over 0<C<1 [47]. In this manner, each of the *N* trajectories in the ensemble carries the same probability, 1/N.

#### 2.2.2. Spin-1/2 Particles

Quantum trajectory formulations for particles with spin 1/2 have also been developed [20,35,36,51,52,53,54,55]. Although it is far from a closed subject, most treatments perform a trace over the spin variables to obtain a single probability density field. Thus, for Hamiltonians that do not depend explicitly on the spin (such as the applications considered here), the time evolution of the density (or even the spatial part of the wavefunction) will proceed the same, regardless of spin state.

Interestingly, the same cannot be said of the quantum trajectories themselves—even though they preserve probability, and, thus, in 1D would be uniquely determined by the density, as discussed in Section 2.2.1. For spin-1/2 particles, the spin state influences the velocity field, and thereby the quantum trajectory dynamics, even for systems with identical density evolutions. However, paradox is averted by the fact that for particles with spin, the system dimensionality per particle is inherently 3D rather than 1D.

The derivation of the quantum trajectory formulation for particles with spin 1/2 is rather involved, so in the interest of space (and indeed, as in the spin-free case), we forego a detailed discussion here, and instead refer the reader to the articles cited above. We do, however, provide an expression for the velocity field $v=\dot{r}$, as we shall avail ourselves of the shortcuts described in Section 2.2.1 for computational work.

Let $\vec{\Psi }$ denote the spinor wavefunction (two complex components) for a single spin-1/2 particle, factored into spatial and spin contributions as follows:(13)$$\vec{\Psi }\left(r,t\right)=\psi \left(r,t\right)\vec{\chi }\left(r,t\right)$$
In Equation (Equation 13) above, r=(x,y,z) is the position vector in 3D physical space, ψ represents the spatial part of the wavefunction as before, and $\vec{\chi }$ represents a Bloch spinor that is normalized to unity $\left({\vec{\chi }}^{†}\vec{\chi }=1\right)$.

In general, the velocity field is given by
(14)v(r,t)=ℏmImΨ→†∇Ψ→Ψ→†Ψ→+ℏ2m∇×(Ψ→†σ→Ψ→)Ψ→†Ψ→,
where $\vec{\sigma }$ is the Pauli spin matrix vector. Since we are considering systems where the Hamiltonian does not explicitly depend on spin, $\vec{\chi }$ is constant. Because of this, Equation (Equation 14) can be simplified to
(15)v(r,t)=ℏmIm∇ψψ+1m∇(ln|ψ|2)×s
where s is the spin vector associated with the Bloch spinor:(16)$$s=\frac{\hbar }{2}{\vec{\chi }}^{†}\vec{\sigma }\vec{\chi }.$$

#### 2.2.3. Dwell Times

In the computation of time spent within the interval $\left[x_{L},x_{R}\right]$, it is only natural that quantum trajectories be considered. In particular, each quantum trajectory x(t), during the course of its evolution from t→−∞ to t→+∞, spends a precisely determined amount of time within the interval—which we call the quantum trajectory *traversal time*, $\tau^{q}$. Moreover, in the time-dependent wavepacket context, the x(C,t) ensemble provides a ready-built distribution, across which statistical averages for $\tau_{C}^{q}$ (and other trajectory-based quantities) may be easily computed.

Let us first, however, consider the even more straightforward case of 1D time-*independent* scattering applications. Here, it is well established that the ensemble reduces to a *single* quantum trajectory [42,44]—which is, moreover, always moving with either positive ($\dot{x}>0$) or negative ($\dot{x}<0$) velocity, depending on if $\phi_{k}\left(x\right)$ is a left-incident (k>0) or right-incident (k<0) scattering solution. Thus, for each $\phi_{k}\left(x\right)$ solution, there is a single well-defined traversal time quantity, $\tau_{k}^{q}$.

It is natural to imagine that $\tau_{k}^{q}=\tau_{k}$, but, in fact, this is *not* the case in general. What we instead have is the relation [5,6],
(17)$$j={\left|\psi \left(x,t\right)\right|}^{2}\dot{x},$$
which implies
(18)$$\tau_{k}=\left(\frac{j}{j_{\mathrm{in}\;}}\right)\tau_{k}^{q}=P_{T}\left(k\right)\tau_{k}^{q}.$$
Thus, in general, the actual quantum dwell time is substantially *lower* than the corresponding quantum trajectory traversal time.

It is worth examining this situation in some detail for the insight it provides. As a rule, dwell times tend to be on the order of the *classical* trajectory traversal time $\tau_{k}^{c}$—which, in the limit that $x_{L}\to -\infty$, $x_{L}\to \infty$ or V(x)→0, becomes $\tau_{k}^{c}\to \left(x_{R}-x_{L}\right)m/\hbar k$. Quantum trajectory dwell times tend to do the same, but only in the absence of interference. Thus (for k>0 solutions), the x→∞ part of the unipolar quantum trajectory travels with classical velocity ℏk/m, but the x→−∞ part is subject to interference between the incident and reflected waves. As a result, in addition to being highly oscillatory, the quantum trajectory in this region moves at a (sometimes *much*) slower average speed—commensurate with flux *j* rather than $j_{\mathrm{in}}$. This situation is examined more closely in Section 2.2.4 and Section 3.1.

Interestingly, the same conclusions do *not* hold for time-dependent wavepackets—at least not those that satisfy the rigorous left-incident boundary conditions, as discussed in Section 2.1.2. More precisely, one tends to find $\tau_{C}^{q}$ values that are comparable to their classical counterparts in *both* asymptotic regions. Thus, provided that the interval limits are sufficiently large, $\tau_{C}^{q}$, $\tau^{c}$, and τ values may all be expected to be comparable. The reason is that discussed at the end of Section 2.1.2: for wavepackets, incident and reflected waves are temporally separated, and, therefore, do not interfere. Moreover, that the asymptotic wavepacket components move with roughly classical speeds is well established, e.g., using semiclassical and/or dispersion-relation arguments [17,56].

In any event, given our distribution of quantum trajectory traversal times $\tau_{C}^{q}$, across the uniform ensemble of trajectories x(C,t), it is natural to define the mean (or distribution-averaged) quantum trajectory traversal time for wavepackets as
(19)$$\tau^{q}=\int_{0}^{1}\;\tau_{C}^{q}\;dC$$
(Note: in a sense, Equation (Equation 19) also applies to time-independent quantum trajectory ensembles, except that there, all trajectories are the same, and therefore have identical $\tau_{C}^{q}$ values.) Once again, we are prompted to ask the question, “Does $\tau^{q}=\tau$?”. This time, the answer is “yes”, as will be demonstrated below.

Consider applying Equation (Equation 12) twice, once for $x=x_{L}$, and once for $x=x_{R}$. This results in expressions for $C_{L}\left(t\right)$ and $C_{R}\left(t\right)$, which label the trajectories that are crossing the left and right interval boundaries, respectively, at time *t*. Note that since the trajectories are moving, these values vary over time. Next, we observe that
(20)$$C_{R}\left(t\right)-C_{L}\left(t\right)=\int_{x_{L}}^{x_{R}}{\left|\psi \left(x,t\right)\right|}^{2}\;dx=\int_{C_{L}}^{C_{R}}dC.$$
Thus, the *x* integral in Equation (Equation 7) is converted into a simpler integral in *C* space, resulting in
(21)$$\tau =\int_{-\infty }^{\infty }\left(\int_{C_{L}}^{C_{R}}dC\right)dt$$
Any *C* value within the range $\left[C_{L}\left(t\right),C_{R}\left(t\right)\right]$ corresponds to a quantum trajectory that lies within the dwell time interval $\left[x_{L},x_{R}\right]$ at time *t*. Thus, for a given fixed *C* value, the time integral in Equation (Equation 21) represents the total amount of time that that particular trajectory spends within the interval—i.e., $\tau_{C}^{q}$. Equation (Equation 21) thus becomes $\tau =\int_{0}^{1}\tau_{C}^{q}\;dC=\tau^{q}$.

#### 2.2.4. Bipolar Quantum Trajectories

The unipolar quantum trajectories described thus far in this section act like classical trajectories when there is no interference, but behave completely differently in the presence of wave interference. In particular, oscillations in the density due to constructive and destructive interference imply (via Equation (Equation 17)) oscillating trajectories—together with an average velocity that can be a tiny fraction of the classical *v* value, when interference is severe. In comparison with classical or semiclassical mechanics, we learn that wave interference is usually associated with *multivalued* velocity fields—i.e., more than one trajectory passing through each point. If, then, a suitable bipolar wave decomposition of the form of Equation (Equation 1) could be found such that the components $\psi_{\pm }\left(x,t\right)$ were themselves interference-free, the corresponding bipolar quantum trajectories would be nonoscillatory and otherwise classical-like everywhere. Such is the idea behind the bipolar approach [27,28,29,30,31,32].

Bipolar quantum trajectories may be especially relevant for quantum dwell times, given that the latter tend to correlate pretty closely with classical traversal times. Indeed, the quantum traversal times $\tau_{k}^{q}$, as exhibited by unipolar quantum trajectories, can be *much* larger than their dwell time counterparts—*precisely* when interference is substantial, as implied by Equation (Equation 18). Would it be possible, therefore, to define a similar relation for $\tau_{k}$ in terms of the *bipolar* trajectory traversal times, $\tau_{k}^{\pm }$—but of a more direct form, and for which $\tau_{k}^{\pm }$ values are comparable to $\tau_{k}$ (and presumably $\tau_{k}^{c}$), even when interference is severe? In this section, we take a first look at this intriguing scenario.

In actuality, what may prove to be even more intriguing is the *converse* relationship—i.e., what dwell times may be able to tell us about bipolar decompositions of the Equation (Equation 1) form. In fact, there are several different bipolar decomposition schemes that have been developed, which work quite well in practice, under a variety of circumstances. However, no single scheme developed to date is perfect, nor can one be singled out as the most theoretically compelling. The quantum dwell time may well provide such an avenue. The idea would be to compute the bipolar velocity field across different *x* values, by computing $\tau_{k}$ as a function of $x_{L}$ say (for fixed $x_{R}$, or vice versa).

Of course, one difficulty with the above scheme is the fact that $\tau_{k}$—although comparable in magnitude to the classical traversal time $\tau_{k}^{c}$—still oscillates with $x_{L}$ or $x_{R}$. Since the interference-free bipolar dwell times do not oscillate, this suggests a correspondence with $\langle \tau_{k}\rangle$ (defined as in Section 2.1.1), rather than with $\tau_{k}$ itself. In actuality, in many respects, this is preferred; the relation between $\tau_{k}$ and other quantum time quantities, such as Smith lifetimes and time delays, invariably involves a *removal* of the oscillatory contribution—effectively replacing $\tau_{k}$ with $\langle \tau_{k}\rangle$. Asymptotically, therefore, we expect no difficulties, although it is not clear whether this process can be extended to intervals with $x_{L}$ or $x_{R}$ in the interaction region of the scattering potential.

All of the above said, for free-particle Hamiltonians (i.e., V(x)=0), as are considered here, there is no ambiguity as to how to effect the bipolar decomposition—and thereby compute bipolar quantum trajectories. For time-independent solutions,
(22)ϕk(x)=exp(ikx)2π=ϕk+(x)for k>0ϕk−(x)for k<0
Thus, each plane-wave solution is a pure $\phi_{k}^{\pm }\left(x\right)$ component, depending on the sign of *k*.

Likewise for time-dependent wavepacket decompositions as per Equation (Equation 1), at any time *t*, the bipolar contributions are obtained as the positive and negative Fourier contributions, respectively, i.e.:(23)$$\psi_{\pm }\left(x,t\right)=\int_{0}^{\pm \infty }\;\langle \phi_{k}|\psi_{0}\rangle \exp\left(-i\hbar k^{2}t/2m\right)\phi_{k}\left(x\right)\;dk$$
Left-incident free-particle solutions thus satisfy $\psi \left(x,t\right)=\psi^{+}\left(x,t\right)$ at *all* times *t*, not only as t→−∞. The same is true for the right-incident solutions, for which $\psi \left(x,t\right)=\psi^{-}\left(x,t\right)$ for all *t*.

The above equality of ψ with $\psi^{\pm }$ makes sense, given that there is no scattering. However, it implies that there is no difference between unipolar and bipolar waves, if rigorous left/right-incident boundary conditions are imposed. What if such conditions are *not* imposed on our wavepacket dynamics? This situation is reconsidered in Section 3, as it characterizes all of the applications considered in this work.

One other attribute of the present applications merits a final comment. In Equation (Equation 22) above, $\phi_{k=0}\left(x\right)$ does not appear. In fact, this is because of an additional restriction on the form of ψ(x,t), which is necessary in order to ensure that the dwell time does not diverge. This requires (for free-particle systems) that $\langle \phi_{k}|\psi_{0}\rangle \to 0$ as k→0 [7,57]. This condition is satisfied for all applications considered here, but would not be true, e.g., for a stationary dispersing Gaussian wavepacket.

### 2.3. Dwell Time Distributions

The quantities referred to here as the dwell times—i.e., τ and $\tau_{k}$—can be regarded as the mean values of *distributions* of various types. It is worth exploring the differences among these distributions in detail—in part because some authors [7,36] have suggested that experiments may be capable of measuring not only dwell time distribution first moments (i.e., the dwell times themselves) but also higher-order moments.

To begin with, there are the distributions that stem from the dwell time operator $\hat{T}$, as discussed in Section 2.1.2. For time-dependent wavepackets, this is defined as [7]
(24)$$\rho_{\hat{T}}\left(\tau \right)=\langle \psi |\delta \left(\hat{T}-\tau \right)|\psi \rangle .$$
Interestingly, even the time-independent version has a nontrivial distribution, across two time values whose average value is $\tau_{k}$. This is because $[\hat{H},\hat{T}]=0$, and so the two time values are eigenvalues of the 2×2$\hat{T}$ matrix, as represented in the $\phi_{k}^{\pm }\left(x\right)$ basis.

There is also $\rho_{\mathrm{FFCF}}\left(\tau \right)$, the distribution based on the FFCF approach discussed in Section 1, which relies on the quantum mechanical flux operator. This approach will not be described in detail here; the interested reader is directed to Refs. [7,41]. Suffice to say that here, too, there are both time-dependent and time-independent versions, with even the latter providing a distribution across a range of τ values.

Finally, we have the (unipolar) quantum trajectory distribution, $\rho_{q}\left(\tau \right)$. For time-dependent wavepackets, this can be derived from the quantum trajectory traversal times across the ensemble—i.e., the $\tau_{C}^{q}$. Note that for time-independent applications (at least in 1D), since there is only one quantum trajectory, there is but a single distribution value—i.e., $\rho_{k}^{q}\left(\tau \right)=\delta \left(\tau -\tau_{k}\right)$.

In principle—and certainly in the time-independent case—these three distributions are different from each other. Yet, they all yield the same first moment—i.e., the mean dwell time itself, τ or $\tau_{k}$. Interestingly, it has also been shown that second moments agree between $\rho_{\hat{T}}\left(\tau \right)$ and $\rho_{\mathrm{FFCF}}\left(\tau \right)$ [7]—though not with $\rho_{q}\left(\tau \right)$, at least in the time-independent case. Higher moments are certainly different across the different distributions, however, thereby presenting the prospect of possible experimental validation.

In a somewhat different category, we also have the bipolar quantum trajectory traversal times. In principle, we have both $\tau_{k}^{+}$ and $\tau_{k}^{-}$ values for a given time-independent solution, although in the present free-particle context, at least, these are the same. Again, the $\tau_{k}^{\pm }$ relate not to $\tau_{k}$ itself, but to the average dwell time $\langle \tau_{k}\rangle$, as will be discussed in Section 3.1. As for the time-dependent bipolar distributions, these are discussed in Section 3.2.1 and Section 3.3.

## 3. Free-Particle Applications

### 3.1. Plane-Wave Superposition States

Consider the most general possible time-independent superposition state of energy $E=\hbar^{2}k^{2}/2m$, described by Equation (Equation 4). Next, imagine that the scattering potential lies completely outside of the desired interval, $\left[x_{L},x_{R}\right]$, either to the right or left of it. Then, within the interval, the individual $\phi_{k}\left(x\right)$’s behave as free-particle bipolar plane-wave components, as per Equation (Equation 22). We therefore posit that *within the interval, there is no difference between a left- (or right-) incident scattering solution $\phi_{k}\left(x\right)$, and a global free-particle superposition state $\psi_{k}\left(x\right)$*. Whether the interference is global, or is caused by an external scattering center, is immaterial.

The above supposition provides us with what we need to define (time-independent) dwell times for superposition states, at least in potential-free regions. We thus find that, as per the discussion in Section 2.1.1, Equation (Equation 3) indeed still holds—with $\phi_{k}\left(x\right)\to \psi_{k}\left(x\right)$, and with the appropriate $j_{\mathrm{in}}$ determined by which constant is larger, $|A_{+}|$ or $|A_{-}|$. Note that $|A_{+}\left|>\right|A_{-}|$ corresponds to the scattering case where the scattering center is to the right of the interval (i.e., $x>x_{R}$), so that $|A_{+}|\phi_{k}^{+}\left(x\right)$ corresponds to the (left-)incident wave. Conversely, $|A_{+}\left|<\right|A_{-}|$ corresponds to a scattering center situated at $x<x_{L}$, with $|A_{-}|\phi_{k}^{-}\left(x\right)$ analogous to the (right-)incident wave. Note that for $|A_{+}\left|=\right|A_{-}|$, we cannot tell which wave is which; thus, the computed dwell time $\tau_{k}$ should be the same whether the solution is presumed to be left-incident or right-incident.

In any event, we now have a method for computing dwell times for superposition states, at least in potential-free regions. From Equations (Equation 4) and (Equation 22), we obtain the following expression for the density:(25)$$|\psi_{k}{\left(x\right)|}^{2}=\frac{|A_{+}{|}^{2}+|A_{-}{|}^{2}+2|A_{+}\left|\right|A_{-}|\cos\left(2kx+\phi_{+}-\phi_{-}\right)}{2\pi },$$
where $\phi_{\pm }=\arg\left(A_{\pm }\right)$. Integrating Equation (Equation 25) over the interval, $\left[x_{L},x_{R}\right]$, and dividing by
(26)$$j_{\mathrm{in}}=\frac{\max(|A_{+}{|}^{2},|A_{-}{|}^{2})}{2\pi }\left(\frac{\hbar k}{m}\right),\;\mathrm{we}\;\mathrm{obtain}$$
(27)$$\tau_{k}=\frac{mL}{\hbar k}\left\{\left[\frac{|A_{+}{|}^{2}+{\left|A_{-}\right|}^{2}}{|A_{\max}{|}^{2}}\right]+\frac{2}{kL}\left[\frac{|A_{+}\left|\right|A_{-}|}{|A_{\max}{|}^{2}}\right]\cos\left[k\left(x_{L}+x_{R}\right)+\phi_{+}-\phi_{-}\right]\sin\left(kL\right)\right\},$$
where $L=(x_{R}-x_{L})$, and $|A_{\max}{|}^{2}=\max(|A_{+}{|}^{2},|A_{-}{|}^{2})$.

Clearly, the $\tau_{k}$ expression of Equation (Equation 27) above is symmetric with exchange of $|A_{+}|$ and $|A_{-}|$, and also oscillates with variations of either $x_{L}$ or $x_{R}$. Since the second term on the right-hand side is the oscillatory contribution, removing it results in a compact expression for the *average* dwell time (in the sense of Section 2.1.1), i.e.,
(28)$$\langle \tau_{k}\rangle =\frac{mL}{\hbar k}\left[\frac{|A_{+}{|}^{2}+{\left|A_{-}\right|}^{2}}{|A_{\max}{|}^{2}}\right]=\frac{mL}{\hbar k}\left[1+P_{R}\left(k\right)\right],$$
where $P_{R}\left(k\right)=1-P_{T}\left(k\right)$ is the “reflection probability” due to a hypothetical scattering center, lying either to the left or right of the interval. Note that k>0 throughout this subsection; thus, all dwell time quantities above are positive.

Let us now consider the quantum trajectory traversal times, starting with unipolar quantum trajectories. Our original supposition from the start of this subsection applies to quantum trajectories as well, and in effect implies that Equation (Equation 18) still holds. In terms of amplitudes, this becomes
(29)$$\tau_{k}=\frac{\left||A_{+}{|}^{2}-{\left|A_{-}\right|}^{2}\right|}{|A_{\max}{|}^{2}}\tau_{k}^{q}.$$
Thus, as before, the unipolar quantum trajectory traversal time can be much longer than the dwell time, when interference between left- and right-incident waves is significant. However, $\tau_{k}$ and $\tau_{k}^{q}$ are in “lock-step” with each other, with respect to their oscillations with $x_{L}$ or $x_{R}$.

Next, let us consider the bipolar quantum trajectories—which, as already discussed in Section 2.2.4, we expect to be affiliated with $\langle \tau_{k}\rangle$, rather than $\tau_{k}$. Since the bipolar components are simple plane waves, their individual traversal times must be equal to the classical time, i.e., $\tau_{k}^{\pm }=\tau_{k}^{c}=mL/\hbar k$. Equation (Equation 28) then implies the simple and eminently sensible result,
(30)$$\langle \tau_{k}\rangle =\tau_{k}^{\pm }+P_{R}\left(k\right)\;\tau_{k}^{\mp }.$$
Thus, the average dwell time is the sum of contributions from both bipolar wave components. The first is a “full strength” contribution from the incident wave, as it passes through the interval. Then, after a hypothetical reflection, the particle once again passes through the same interval in the opposite direction, making a second contribution to the average dwell time—although this time weighted by the reflection probability. A similar analysis was considered in Ref. [7], albeit only for $P_{R}\left(k\right)=1$.

### 3.2. Bipolar Superposition/Decomposition for Free-Particle Wavepackets

#### 3.2.1. Definition of Dwell Time Quantities

For the free-particle Hamiltonian case at least, there are several ways to arrive at unambiguous expressions for the bipolar dwell times $\tau_{\pm }$, and for the average dwell time $\langle \tau \rangle$. The most straightforward is to substitute Equation (Equation 1) into Equation (Equation 7), to obtain
(31)τ=∫−∞∞∫xLxRψ+(x,t)2dxdt+∫−∞∞∫xLxRψ−(x,t)2dxdt+…=τ++τ−+2Re∫−∞∞∫xLxRψ−*(x,t)ψ+(x,t)dxdt.
Note that whereas the integrands comprising $\tau_{\pm }$ above are positive definite, the last term on the second line above has a highly oscillatory integrand with respect to *x*.

The above state of affairs highly suggests the following as the *definition* of the average dwell time:(32)$$\begin{array}{cc} \langle \tau \rangle & =\tau_{+}+\tau_{-}\\ \mathrm{with}\;\tau_{\pm } & =\int_{-\infty }^{\infty }\left(\int_{x_{L}}^{x_{R}}{\left|\psi_{\pm }\left(x,t\right)\right|}^{2}dx\right)dt \end{array}$$
Note that we are here using bipolar dwell times to obtain the average dwell time, which works because the bipolar decomposition for free-particle systems is unambiguously defined via Equations (Equation 22) and (Equation 23). Going forward, we may well adopt the converse tack, whereby average dwell times, if they can be unambiguously defined throughout the scattering region (which may be suggested by earlier work [6]), are used to define the $\tau_{\pm }$.

In any event, it is clear that the above definitions are eminently sensible. In particular, $\langle \tau \rangle$ and $\tau_{\pm }$ are nonoscillatory with respect to $x_{L}$ and $x_{R}$, provided that the same is true of $\psi_{\pm }\left(x,t\right)$, which is expected. Also, it is clear that the final, oscillatory integral in Equation (Equation 31) vanishes (so that $\langle \tau \rangle \to \tau$) in any of the following limits:Broad $\psi_{\pm }$ wavepackets (i.e., narrow wavepackets in Fourier space);Wide intervals [large $L=(x_{R}-x_{R})$], regardless of location;Aasymptotically located intervals, regardless of width.

Note that $\psi_{\pm }\left(x,t\right)$ describe left- and right-moving, preferably localized wavepackets, that necessarily cross each other in some region of spacetime—which, without loss of generality, we may take to be centered at (x,t)=(0,0). When this occurs, the two bipolar wavepacket components interfere, potentially causing oscillations in the unipolar density ${\left|\psi \left(x,t\right)\right|}^{2}$ and/or oscillations and reflections in the unipolar quantum trajectories. “Asymptotically located intervals” are, thus, those that lie well outside the “collision region” of the bipolar wavepacket components—i.e., with either $x_{L}>0$ or $x_{R}<0$ large compared to the collision width. At asymptotically large times, we expect ψ(x,t) to comprise spatially well-separated lobes—with the (x→−∞) and (x→+∞) lobes corresponding to $\psi_{+}$ and $\psi_{-}$, respectively, for t→−∞, and with these associations reversed for t→+∞.

#### 3.2.2. Probability Weights

That $\langle \tau \rangle$ is the *sum* of $\tau_{+}$ and $\tau_{-}$, and not their average, simply reflects the fact that the bipolar waves are not normalized to unity. Instead, we have
(33)$$p_{\pm }=\langle \psi_{\pm }|\psi_{\pm }\rangle =\int_{-\infty }^{\infty }\;{\left|\psi_{\pm }\left(x,t\right)\right|}^{2}\;dx$$
with $p_{+}+p_{-}=1$, since $\langle \psi_{+}|\psi_{-}\rangle =0$ (at least for free-particle wavepackets, as per Equations (Equation 22) and (Equation 23)). Thus, the $p_{\pm }$ values represent bipolar *probability weights*, which remain constant over time, and are also built into the definition of the $\tau_{\pm }$. As a consequence, Equation (Equation 32) in effect represents a probability-weighted average.

Note that this interpretation is entirely consistent with the time-independent form of Equation (Equation 30), for which the $\tau_{k}^{\pm }$ quantity is *not* probabilistically weighted, and therefore the $P_{R}\left(k\right)$ factor must be introduced explicitly. In any event, it is clear from the definitions that the bipolar time-independent and time-dependent dwell time quantities enjoy the same simple probability-weighted relationship as do the corresponding unipolar quantities, as expressed in Equation (Equation 10). We thus have
(34)$$\tau_{\pm }=\int_{0}^{\infty }\;{\left|\langle \phi_{\pm k}|\psi_{0}\rangle \right|}^{2}\tau_{k}^{\pm }\;dk.$$
Equation (Equation 34) may be regarded as an alternate definition of the $\tau_{\pm }$, that holds at least for free-particle systems.

Given that the $\tau_{\pm }$ definitions already include their probabilistic weights $p_{i}$, this fact has ramifications for the proper treatment of bipolar quantum trajectories. Generally speaking, these are handled in fully analogous fashion to the unipolar quantum trajectories. Note that each $\psi_{\pm }$ component has *its own* ensemble of bipolar quantum trajectories, labeled, respectively, by the trajectory labeling coordinates, $0<C_{\pm }<1$. The trajectories belonging to a given $C_{\pm }$ ensemble do not cross each other, but, of course, the $C_{+}$ and $C_{-}$ trajectories do cross—with one of each type passing through every spacetime point, (x,t).

From Equation (Equation 19), we see that for a numerical simulation comprising *N* uniformly-distributed (in *C* space) unipolar quantum trajectories, each such trajectory carries an equal probability of 1/N. Thus, $\tau =\tau^{q}$ is computed by simply summing over all of the *N* individual quantum trajectory traversal times, and then multiplying by 1/N. For bipolar quantum trajectories, a similar procedure is followed, except that each bipolar quantum trajectory only carries a probability of $p_{\pm }/N_{\pm }$, with $N_{\pm }$ being the number of trajectories in the corresponding bipolar ensemble.

#### 3.2.3. Symmetric Example

Consider a wavepacket that is real-valued and odd-symmetric at t=0, i.e., $\psi_{0}\left(-x\right)=-\psi_{0}\left(x\right)$. Since $\psi_{0}$ is real-valued, Equation (Equation 11) is zero everywhere, so that the wavepacket is momentarily stationary. The time evolution of ψ(x,t) is symmetric in time, exhibiting no net motion, but dispersing outwards symmetrically in *x* with increasing |t|. In particular, ψ(−x,t)=−ψ(x,t), and $\psi \left(x,-t\right)=\psi {\left(x,t\right)}^{*}$ for all (x,t).

The unipolar quantum trajectories for this system exhibit symmetry in both *x* and *t*—converging towards the “collision region” while t<0, coming to a complete halt at t=0, and then turning around and fanning outwards again for t>0. While in the collision region, the colliding waves interfere, which may cause the unipolar quantum trajectories to oscillate, but such oscillations are expected to dissipate in the asymptotic regions.

Given the reality of $\psi_{0}\left(x\right)$, it is clear that the bipolar waves must satisfy $\psi_{0-}\left(x\right)=\psi_{0+}{\left(x\right)}^{*}$ at t=0. Thus, $|\psi_{0+}{\left(x\right)|}^{2}={\left|\psi_{0-}\left(x\right)\right|}^{2}$, and $p_{\pm }=1/2$. Also, $\psi_{0\pm }{\left(x\right)}^{*}=-\psi_{0\pm }\left(-x\right)$. More generally, the time-evolving bipolar waves satisfy the following symmetry relationships:(35)$$\psi_{-}\left(x,t\right)=-\psi_{+}\left(-x,t\right)=\psi_{+}{\left(x,-t\right)}^{*}$$
Thus, Reψ±(0,t)=0 and ψ(0,t)=0 for all *t*.

Unlike the unipolar quantum trajectories, individual bipolar trajectories for $\psi_{+}$ are always moving to the right (i.e., in the positive *x* direction), whereas those for $\psi_{-}$ are always moving to the left. Also, the bipolar trajectories are (ideally) smooth and well behaved in the sense that they do not exhibit oscillations. The bipolar velocity fields are symmetric in time and space, in that $v_{\pm }\left(-x,-t\right)=v_{\pm }\left(x,t\right)$, and $v_{-}\left(-x,t\right)=-v_{+}\left(x,t\right)$.

Regarding the dwell times, clearly, $\tau_{-}=\tau_{+}$, so that $\langle \tau \rangle =2\tau_{\pm }$. At asymptotic times, ψ(x,t) consists of two lobes, one corresponding to $\psi_{+}\left(x,t\right)$, and the other to $\psi_{-}\left(x,t\right)$. These are symmetric in *x*, but switch roles as *t* changes sign. Consequently, for asymptotic intervals, $\left[x_{L},x_{R}\right]$, there is no interference, and so $\tau =\tau_{+}+\tau_{-}=2\tau_{\pm }$. This can be interpreted as follows. One half of the unipolar trajectories pass through the same asymptotic interval twice; once on their way in to the collision center, and once on their way out. Thus, on average, we have one traversal per unipolar trajectory.

Of the two unipolar traversals, one of these corresponds to a single traversal of the interval by *all* of the $\psi_{+}=\left(x,t\right)$ bipolar trajectories—and, thus, to one traversal per bipolar trajectory, on average. However, these are weighted by $p_{+}=1/2$. Similar comments apply to the $\psi_{-}\left(x,t\right)$ bipolar trajectories, which necessarily correspond to the other unipolar traversal. Hence, $\tau =\tau_{+}+\tau_{-}$. Of course, intervals that lie within (or contain) the collision center need not satisfy this relation.

### 3.3. Spin 1/2: Das and Dürr Application

#### 3.3.1. Introduction

In Refs. [35,36], the authors propose an experimental test of quantum trajectory arrival times for a single spin-1/2 particle trapped in a cylindrical waveguide. The end face of the waveguide blocks access to the z<0 region, whereas the perpendicular potential,
(36)$$V\left(x,y\right)=\frac{1}{2}m\omega^{2}\left(x^{2}+y^{2}\right),$$
models that of a quadrupole ion trap.

In Ref. [36], up until time t=0, the particle is confined in the *z* direction as well, via an additional $\left(1/2\right)mz^{2}$ contribution to the potential. For all t≤0, the spatial part of the wavefunction, i.e., ψ(r,t), is taken to be a stationary eigenstate of Equation (Equation 2). Specifically, we have the first-excited harmonic oscillator state in the *z* direction, and ground state (Gaussian) in the perpendicular directions, i.e.,
(37)$$\psi_{0}\left(r\right)=\left(\frac{\sqrt{4\omega }}{\pi^{3/4}}\right)\exp\;\left[-\frac{\omega }{2}\left(x^{2}+y^{2}\right)\right]\;z\;e^{-z^{2}/2}\;\;z>0$$
Note that an odd excitation in *z* is required, in order to ensure that the wavefunction vanishes as $z\to 0^{+}$, since the z<0 region is not used.

In the Das and Dürr treatment, the *z* potential is suddenly “turned off” at t=0, allowing the formerly stationary wavepacket to disperse outward in *z* along the waveguide as time progresses (t>0), in accordance with the time-dependent Pauli equation [35,36,51]. Note that the spin and spatial components in Equation (Equation 13) are decoupled throughout—with the spinor $\vec{\chi }\left(r,t\right)={\vec{\chi }}_{0}$ remaining constant. On the other hand, the (unipolar) quantum trajectory dynamics depend very much on the spin orientation.

Two distinct cases are considered, which are called “spin up” and “spin up-down”, defined as follows:(38)Ψ→↑(r,0)=ψ0(r)10;Ψ→↕(r,0)=ψ0(r)1/21/2
For ${\vec{\Psi }}_{↑}$, the quantum trajectories turn out to be dynamically independent across *z* and (x,y); but this is not the case for the ${\vec{\Psi }}_{↕}$ trajectories, which are accordingly far more complex (although they are confined to the *y*–*z* plane). In each case, an appropriate ensemble of quantum trajectories is propagated, and used to construct a distribution of arrival times. These differ markedly between the two spin cases—thus motivating the idea that they can be distinguished experimentally, although that notion has been questioned [37,38].

In any event, we take a slightly different approach here. Specifically, our use of the dwell time requires a time-independent Hamiltonian over *all* time, in order for the dwell time operator to be Hermitian [7]. Thus, our Hamiltonian potential is always that of Equation (Equation 36), extended across −∞<t<∞. Our wavepacket density thus disperses outwards in both t>0 and t<0 directions, and is temporarily stationary only at t=0. In this manner, the dwell time, even for intervals close to the z=0 origin, is always found to be finite—unlike in the Das and Dürr treatment, for which the dwell time technically always diverges.

We can demonstrate the finiteness of our dwell times as follows. Consider that our time-dependent spatial wavefunction solution always takes the separable form $\psi \left(r,t\right)=\psi_{xy}\left(x,y\right)\psi_{z}\left(z,t\right)$, regardless of the spinor state. Moreover, $\psi_{z}\left(z,t\right)$ is always the “right half” of the spatially extended solution over −∞<z<∞, with $\psi_{z}\left(-z,t\right)=-\psi_{z}\left(z,t\right)$. Since $\psi_{z}\left(z,t\right)$ describes free-particle evolution, the Fourier transform is the same for all *t*, apart from phase shifts, with the form at t=0 given by Equation (Equation 37). Since this form is that of a harmonic oscillator eigenstate, which is well known to equal its own Fourier transform (apart from scaling), the Fourier function must be linear in *k* in the vicinity of k=0, for all *t*. But this is the necessary condition for finite dwell times, as discussed at the end of Section 2.2.4.

#### 3.3.2. Unipolar Time Evolution

Since the wavepacket evolution is separable, and since the initial perpendicular wavefunction is an eigenstate of the perpendicular Hamiltonian, it remains so for all time. The only interesting wavepacket dynamics, therefore, correspond to $\psi_{z}\left(z,t\right)$. Here, the 1D time evolution is in accordance with the free-particle propagator, which gives rise to dispersion in both time directions as follows:(39)$$\psi_{z}\left(z,t\right)=\frac{2}{\pi^{1/4}}\frac{z}{{\left(1+it\right)}^{3/2}}\exp\left[-\frac{z^{2}}{2(1+it)}\right]$$
Note that the specific values ℏ=m=1 are presumed in Equation (Equation 39) above, and throughout the rest of this work.

The unipolar quantum trajectory dynamics are determined by Equation (Equation 14), which in turn depend on the particular spin state. Below, we present specific results for each of the two cases considered here.



*spin up:*





(40)
$$\dot{x}=-\omega y\;;\;\dot{y}=\omega x\;;\;\dot{z}=\left(\frac{t}{1+t^{2}}\right)z$$





*spin up-down:*



(41)$$\dot{x}=0\;;\;\dot{y}=\frac{1}{z}-\frac{z}{1+t^{2}}\;;\;\dot{z}=\omega y+\left(\frac{t}{1+t^{2}}\right)z$$
Note that the spin-up equations decouple *z* from (x,y), whereas the spin-up-down trajectories remain in a *y*–*z* plane, as claimed earlier in Section 3.3.1.

Using the trajectory evolution equations from the preceding paragraph, we computed a uniformly distributed ensemble of quantum trajectories for each of the two spin cases listed above. For the spin-up case, it was only necessary to consider a 1D ensemble (in $C_{z}$ space) of 1D *z*-component trajectories—since all dwell time intervals of interest are *z* intervals, and this component decouples from the perpendicular components. For the spin-up-down case, however, because of the coupling, a 3D ensemble of 2D trajectories is required. The statistical convergence of the latter is, therefore, not as good as for the former, even though significantly more trajectories were used.

#### 3.3.3. Bipolar Time Evolution

For the bipolar analysis, we consider only the spin-up case, for which the *z* evolution becomes decoupled from (x,y), and is therefore purely 1D. This enables us to effect a bipolar decomposition of the $\psi_{z}\left(z,t\right)$ wavefunction of Equation (Equation 39), using Equation (Equation 23). The resultant bipolar wavefunctions become
(42)$$\begin{array}{cc} \psi_{\pm }\left(z,t\right)= & \frac{1}{\pi^{1/4}}\frac{z}{{\left(1+it\right)}^{3/2}}\exp\left[-\frac{z^{2}}{2(1+it)}\right]\left\{1\pm i\;\mathrm{erfi}\;\left[\frac{z}{\sqrt{2(1+it)}}\right]\right\}\\ \; & \mp \frac{i\sqrt{2}}{\pi^{3/4}\left(1+it\right)}\;, \end{array}$$
normalized such that $\int_{-\infty }^{\infty }{\left|\psi_{\pm }\left(z,t\right)\right|}^{2}\;dz=1$ for all *t* (more on this convention below). From the form of Equation (Equation 42) above, it is evident that $\psi_{+}\left(z,t\right)+\psi_{-}\left(z,t\right)$ matches $\psi_{z}\left(z,t\right)$ as given in Equation (Equation 39).

The bipolar wavefunctions $\psi_{\pm }\left(z,t\right)$ as defined above correspond to the symmetric wavepacket example of Section 3.2.3. Note, for instance, that the corresponding unipolar wavepacket of Equation (Equation 39) is odd symmetric in *z*, as required. There is, however, one important difference here, which is that $\psi_{z}\left(z,t\right)$ is set to zero in the range z<0.

This changes things in some interesting ways. First, the unipolar dwell time τ is *double* what it would be in Section 3.2.3. This can be understood either in wave terms (i.e., because of Equation (Equation 7), and the fact that the density $|\psi_{z}{|}^{2}$ is now twice what it would be if the z<0 region were included), or in trajectory terms (i.e., because *all* of the unipolar trajectories now traverse the interval twice). In any event, we end up with two traversals per unipolar trajectory on average, instead of one.

The bipolar analysis is decidedly more interesting. Unlike the unipolar wave, which vanishes as z→0 and, thus, remains confined to the z>0 region, the bipolar waves $\psi_{\pm }\left(z\right)$ move into ($\psi_{+}$) or out of ($\psi_{-}$) this region over time, maintaining positive and negative flux, respectively, at z=0 and across all times. One can thus imagine these solutions extended into the z<0 region, although it is only the z>0 region that contributes probabilistically. As a consequence, the probability weights $p_{\pm }$ are thus now *changing over time*, with $p_{+}\to 0$ or 1, as t→−∞ or +∞ (and vice versa for $p_{-}$).

For asymptotic intervals, $p_{+}\to 1$ by the time that the unipolar wave is passing through the interval at large t>0; thus, all of the $\psi_{+}$ bipolar trajectories pass through the interval once, with a full weight of $p_{+}=1$—resulting in double the $\tau_{+}$ value that would be observed in Section 3.2.3. Of course, similar comments apply to $\tau_{-}$, and the first unipolar traversal of the interval, at large t<0. Ultimately, $\tau =\tau_{+}+\tau_{-}=2\tau_{\pm }$ as before, but all dwell times are doubled.

For arbitrary intervals, $\left[z_{L},z_{R}\right]$ (with $z_{L}\geq 0$) we find that $\langle \tau \rangle =\tau_{+}+\tau_{-}$, and also $\tau_{+}=\tau_{-}$. Note that it is not only the dwell times $\tau_{\pm }$ themselves that are identical, but also the dwell time *distributions*, $\rho_{\pm }\left(\tau \right)$. Indeed, in the special case of asymptotic intervals, these distributions are also equivalent to the unipolar dwell time distribution, $\rho_{q}\left(\tau \right)$—apart from a trivial factor-of-two rescaling of τ.

There is, however, an interesting and natural reinterpretation of the bipolar scenario that will lead to a *different* dwell time distribution for asymptotic intervals (albeit the same mean dwell time, τ). Consider the first of the Equation (Equation 35) relations, which relates, at any given time *t*, the *actual*$\psi_{-}\left(z,t\right)$ wave in the z>0 region, to the “virtual” $\psi_{+}\left(z,t\right)$ wave in the z<0 region, and vice versa. Given this association, it is natural to discard the virtual waves altogether, and instead regard z=0 as a V→∞ scattering center that reflects the outgoing $\psi_{-}$ wave into the incoming $\psi_{+}$ wave.

With this new interpretation, there is a single bipolar wave, denoted ‘$\psi^{\pm }\left(z,t\right)$’ (note the use of superscripts), that is reflected at z=0, and maintains unit probability weighting $p^{\pm }=1$ throughout. The mean dwell time $\tau^{\pm }$ as computed using the $\psi^{\pm }\left(z,t\right)$ is necessarily the same as before—i.e., 〈τ〉. However, the distributions $\rho^{\pm }\left(2\tau \right)$ and $\rho_{\pm }\left(\tau \right)$ are now significantly different. The reason is because there is now just a single ensemble of trajectories, $C^{\pm }$, rather than two ensembles with different trajectories, $C_{\pm }$, that are considered to be uncorrelated. In effect, $\psi^{\pm }\left(z,t\right)$ induces a correlation between $C_{+}$ and $C_{-}$ trajectories. Moreover, for asymptotic intervals, it does so in a way that results in a significantly *narrower* distribution for $\rho^{\pm }\left(\tau \right)$.

To simulate the $\rho^{\pm }\left(\tau \right)$ distribution numerically, the easiest way to achieve this is to work with just $\psi_{+}\left(z,t\right)$, but extended across the entire −∞<z<∞ range. Then, in addition to the original interval over $\left[z_{L},z_{R}\right]$, a second interval is added over $\left[-z_{R},-z_{L}\right]$, to simulate the first traversal at large negative times. The total quantum trajectory traversal time $\tau_{C}^{\pm }$ for a given trajectory *C* is then taken to be the sum of the times required to traverse both intervals. When integrated (or in numerical practice, summed) over *C* as per Equation (Equation 19), the resultant $\tau^{\pm }$ must equal the average unipolar dwell time 〈τ〉, although the correlations will induce a decidedly different dwell time distribution, as discussed.

## 4. Results

### 4.1. Overview

A large number and variety of quantum trajectory calculations were performed for the 3D spin-1/2 system of Das and Dürr [36], as discussed in Section 2.2.2 and Section 3.3. Parameters of this system were chosen as follows: ℏ=1; m=1; ω=5. Both the spin-up and spin-up-down cases were considered, although many more results are reported here for the spin-up case. The reason is that in this case, the *z*-component of the quantum trajectory dynamics separates out from (x,y). This presents at least two advantages. First, accurate 3D statistics may be gathered from a 1D rather than 3D ensemble of trajectories, as discussed. Second, a bipolar treatment in *z* becomes straightforward. We thus report bipolar quantum trajectory results only for the spin-up case, although at least some unipolar results are presented for both spin cases. Unless otherwise indicated, figures and tables presented here refer to the spin-up system only.

We also considered three very distinct interval windows: [10, 20]; [0, 4]; [0, 0.4]. The first is an asymptotic window, situated far beyond the t=0 wavepacket. The second interval, in contrast, begins at $x_{L}=0$, and includes the entire “collision region” (i.e., it contains all significant density, $|\psi_{0z}{\left(z\right)|}^{2}$, at t=0). The third window represents a narrow slice, well in the interior of the collision region, also with $x_{L}=0$. This set of interval windows provides a representative sampling of the types of dwell time behaviors that one may expect to observe in practice. In any event, for the numerical results presented here, all three windows were investigated in the spin-up case, whereas only the first, asymptotic window was considered in the spin-up-down case.

Before computing dwell time quantities, as a “calibration” test, we first reproduced the arrival time calculations of Das and Dürr [36], just to ensure that our numerical calculations were working properly. We achieved near-perfect agreement with their arrival time results, both for the spin-up and the spin-up-down case. For the dwell time calculations themselves, we first computed dwell time distributions, gathered from traversal time statistics for the individual quantum trajectories comprising each type of ensemble. Three distinct types of dwell time distributions were thus obtained, i.e., unipolar [$\rho_{q}\left(\tau \right)$]; bipolar [$\rho_{\pm }\left(\tau \right)$]; bipolar reflected [$\rho^{\pm }\left(\tau \right)$]. In order to ensure numerical convergence of the results, these calculations were repeated over a wide range of (1D) ensemble sizes, up to a maximum size of N=1000 quantum trajectories.

Since the dwell time distributions themselves are evidently important, we shall report on those, as well as on the corresponding dwell time quantities that may be derived from them. The latter include the first moments or mean dwell times, i.e., τ [from $\rho_{q}\left(\tau \right)$], $\tau_{\pm }$ and $\langle \tau \rangle$ [from $\rho_{\pm }\left(\tau \right)$], and $\tau^{\pm }$ [from $\rho^{\pm }\left(\tau \right)$]. In the case of the spin-up $\rho_{q}\left(\tau \right)$ and $\rho^{\pm }\left(\tau \right)$ distributions, we also computed second moments, reported here in the form of standard deviations.

### 4.2. Wavefunctions and Quantum Trajectories

Insofar as the presentation of results is concerned, we begin with Figure 1, a plot of the time evolution of the *z* component wavepacket density, $|\psi_{z}{\left(z,t\right)|}^{2}$, which is common to both the spin-up and spin-up-down applications. This is represented by the gray curves at three different time values, t=0,5,10, with later times corresponding to more spread-out distributions. As predicted, the density consists of two equivalent lobes separated by a node at z=0—although in reality, only the z>0 lobe is “real” (the z<0 lobe is “virtual”). Although colliding wavepackets may in principle exhibit substantial interference within/during the collision window in space/time, we note that this is not the case here. Here, because $|\psi_{z}{\left(z\right)|}^{2}$ is a harmonic oscillator eigenstate, it preserves its initial shape over all time, simply spreading out with a width that increases as $\sqrt{1+t^{2}}$.

This presents an interesting situation for the traveling bipolar wave components, $\psi_{+}\left(z,t\right)$ and $\psi_{-}\left(z,t\right)$, whose time-evolving densities are also indicated in Figure 1, using dashed and dotted curves, respectively. At time t=0, these components have maximum density at z=0—i.e., precisely where ψ(z,t) itself vanishes. At t=0, the node at z=0 thus corresponds to interference between equal and opposite colliding waves. Over time, however, as $\psi_{+}\left(z,t\right)$ moves to the right and $\psi_{-}\left(z,t\right)$ moves to the left, these components no longer interfere, and thus come to form the right and left lobes, respectively, of ψ(z,t), as discussed in Section 3.3.3. For brevity, we refrain from presenting plots that show the Re, Im, or arg parts of the above respective wavefunctions, although this behavior can be largely deduced from the quantum trajectory plots, which we present next. In any event, we note that the symmetry relations of Section 3.2.3 have all been verified, including Equation (Equation 35).

Plots of the unipolar quantum trajectories associated with $\psi_{z}\left(z,t\right)$ are indicated in Figure 2, which also highlights the [0, 4] interval window. Here, only the “real” trajectories, corresponding to z>0, are indicated. From the figure, it is clear that the unipolar trajectories first converge towards z≈0 from above, then slow down and stop at time t=0, where they also reach their closest proximity to each other. Thereafter, they change direction and fan outwards towards z→∞, so that mirror symmetry in *t* is achieved. Beyond the collision region indicated, all trajectories become straight lines. Note that despite turning around, all quantum trajectories are smooth and nonoscillatory in the collision region—an atypical situation, due to the solution being a harmonic oscillator eigenstate, as discussed. In any event, this behavior is characteristic of what has been called a “type one” node [27].

In Figure 3, we find the corresponding bipolar quantum trajectory plots, associated with the $\psi_{+}\left(z,t\right)$ bipolar right-traveling wave. The corresponding plots for $\psi_{-}\left(z,t\right)$ bipolar trajectories are identical, apart from a mirror reflection in *z*. Here, both real and virtual contributions are indicated, to highlight the symmetry properties discussed previously. As predicted, these trajectories move monotonically across *z*, without changing direction. They speed up and slow down a bit while moving through the collision region, but are perfectly smooth and well behaved, as expected. In a similar manner to the unipolar trajectories, the bipolar trajectories continue to fan out and move in straight lines beyond the indicated collision region—in fact, they become equal to the unipolar trajectories in these asymptotic limits.

Note that the bipolar trajectories are used to compute both the $\rho_{\pm }\left(\tau \right)$ and the $\rho^{\pm }\left(\tau \right)$ dwell time distributions. The former is computed by integrating the time that each trajectory spends in the real (z>0) interval (e.g., [0, 4]), whereas the latter derives from the time each trajectory spends in *both* real and virtual intervals (e.g., [0, 4] and [−4, 0]). Note from the figure that a trajectory that moves relatively slowly through one interval moves relatively quickly through the other. Hence, while mean dwell times are necessarily the same due to symmetry, the $\rho^{\pm }\left(2\tau \right)$ distribution may generally be expected to be narrower than $\rho_{\pm }\left(\tau \right)$, especially for asymptotic intervals.

### 4.3. Dwell Time Distributions

Moving on to dwell time distributions, Figure 4, Figure 5 and Figure 6, respectively, present the dwell time distributions, $\rho_{q}\left(2\tau \right)$, $\rho_{\pm }\left(\tau \right)$, and $\rho^{\pm }\left(\tau \right)$, for the [10, 20] interval window. Note that the first two distributions are practically identical to each other, as was predicted to be the case for asymptotic intervals. The $\rho^{\pm }\left(\tau \right)$ distribution is radically different, however—also as predicted. In particular, it is much narrower, and strongly peaked on the low-τ end. Nevertheless, computed (mean) dwell times are in very close agreement, as will be discussed in Section 4.5.

Figure 7, Figure 8 and Figure 9, respectively, present the dwell time distributions, $\rho_{q}\left(\tau \right)$, $\rho_{\pm }\left(\tau \right)$, and $\rho^{\pm }\left(\tau \right)$, for the [0, 4] interval window. Now, we find that $\rho_{q}\left(2\tau \right)$ and $\rho_{\pm }\left(\tau \right)$ are very different from each other—with the latter much more narrowly peaked on the low-τ end. This is because the bipolar trajectories are moving much more uniformly in relation to each other than are the unipolar trajectories. For the latter, the trajectories on the left spend a comparatively long time in the region; they must enter the interval window, penetrate close to z=0, and then turn around and go all the way back before exiting. The right-most trajectories, in contrast, do not have nearly as far to travel, and, thus, spend considerably less time in the region (see Figure 2). For this interval window, $\rho_{\pm }\left(\tau \right)$ and $\rho^{\pm }\left(2\tau \right)$ are actually pretty close to each other, although the latter is still narrower, for reasons already discussed.

Finally, Figure 10, Figure 11 and Figure 12, respectively, present the dwell time distributions, $\rho_{q}\left(\tau \right)$, $\rho_{\pm }\left(\tau \right)$, and $\rho^{\pm }\left(\tau \right)$, for the [0, 0.4] interval window. In comparison with other interval windows, the $\rho_{q}\left(\tau \right)$ distribution here is pushed up much more closely against the origin. This is due to the fact that $|\psi_{z}\left(z,t\right)|\to 0$ as z=0, so that relatively few of the trajectories that enter the window spend significant time there. Those trajectories that either just “graze” the window, or penetrate only a short distance, are probabilistically favored. In any event, very few of the unipolar quantum trajectories even enter the window—only 437 out of 10,000, to be precise.

As for the bipolar dwell time distributions, $\rho_{\pm }\left(\tau \right)$, and $\rho^{\pm }\left(2\tau \right)$ are now found to be nearly identical. Indeed, this finding is entirely to be expected in the $x_{R}\to 0$ limit, because the bipolar quantum trajectories become straight lines over the z>0 and z<0 intervals. Thus, for each individual trajectory, $\tau_{C}^{\pm }\to 2\tau_{C\pm }$, and so the distributions are identical apart from the aforementioned factor-of-two rescaling. In any event, like for the other interval windows, the bipolar distributions are heavily weighted towards smaller τ values. They are also more consistently similar to each other than are the unipolar dwell time distributions, across the range of intervals considered.

As a final note, we comment on the fact that the [0, 0.4] distribution figures above appear to indicate a much larger apparent mean dwell time for $\rho_{q}\left(\tau \right)$ than for $\rho_{\pm }\left(\tau \right)$ or $\rho^{\pm }\left(\tau \right)$. In fact, we shall see presently that the opposite is true. The reason for the apparent discrepancy is that only a small fraction of unipolar trajectories actually enter the interval window, whereas *all* of the bipolar trajectories do so.

### 4.4. Unipolar Quantum Trajectories for Spin-Up-Down Case

We would be remiss not to include at least one quantum trajectory plot for the spin-up-down case. Figure 13 presents one such plot, indicating the *z* component of the unipolar quantum trajectories, with initial values $x_{0}=0$ and $y_{0}=-0.136208$. Note that the trajectories do not cross; they only appear to do so, because we are projecting down to the *z* component only. Even so, it is clear that the trajectory behavior is much more complex than for the spin-up case. This is owing to coupling between *y* and *z*, brought about by the spinor components. From Equation (Equation 41), it is clear that the trajectory oscillations evident in Figure 13 are due to the coupling contribution, which depends on the parameter ω, which characterizes the perpendicular (x,y) potential. Thus, changing the value of ω would lead to substantially different (but still quite complex) quantum trajectories.

### 4.5. Dwell Times, Standard Deviations, and Numerical Convergence

Finally, we extract first and second moments from the aforementioned dwell time distributions, in the form of the (mean) dwell times themselves, as well as their corresponding standard deviations. The spin-up results are presented in Table 1 for all three intervals. The results are in exact accordance with our earlier predictions. In particular, for every interval, we find that $\langle \tau \rangle =\tau_{+}+\tau_{-}=\tau^{\pm }$, with agreement to within the level of numerical accuracy achieved in the calculation (to be discussed shortly). Additionally, for the asymptotic interval [10, 20], $\langle \tau \rangle =\tau$, to the same level of numerical accuracy—also as predicted. Despite this agreement, however, the standard deviations for $\rho_{q}\left(\tau \right)$ and $\rho^{\pm }\left(\tau \right)$ are very different, with $\Delta \tau^{\pm }$ significantly smaller than Δτ, as indicated in the last two columns of the second row of the table.

For the middle interval [0, 4], for which the entire collision region is spanned, Table 1 indicates a small but significant difference between τ and $\langle \tau \rangle$. This is reasonable, given that τ values are known to oscillate with $z_{R}$ in this regime (albeit less so here than is typical for other free-particle examples, as discussed). The $\Delta \tau^{\pm }$ value, though still smaller than Δτ, is much closer to Δτ than for the aymptotic interval. This is because the bipolar trajectories are much more fanned out asymptotically, then they are near the origin—implying a much broader asymptotic distribution for $\rho_{\pm }\left(\tau \right)$ than for $\rho^{\pm }\left(2\tau \right)$ (and recalling, also, that $\rho_{q}\left(2\tau \right)\to \rho_{\pm }\left(\tau \right)$ asymptotically).

The last interval, [0, 0.4], is in some ways the most interesting. Here, the trajectories are essentially straight lines, and so there is no difference between $\rho_{\pm }\left(\tau \right)$ and $\rho^{\pm }\left(2\tau \right)$. In fact, all of the bipolar quantities are seen to scale roughly proportionately (i.e., by a factor of 10) down from their [0, 4] interval values. In contrast, individual unipolar quantum trajectories in this interval have much larger traversal times, because this is the region where the unipolar trajectories are barely moving. Nevertheless, because so few of the unipolar trajectories even enter this interval window (because the probability density is so low), the mean dwell times are extremely small—an order of magnitude or so smaller than for the bipolar quantities. Similar comments also apply to Δτ vs. $\Delta \tau^{\pm }$.

Numerical convergence for the unipolar dwell time results may be examined in Table 2, where we also consider the spin-up-down case. In this table, only the asymptotic interval is considered. For each of the two spin cases, the table indicates computed values for τ, as a function of the number of trajectories, *N* (or $N^{3}$, in the spin-up-down case). As expected, the convergence is much faster for the spin-up system (in terms of the total number of trajectories), because the distribution of trajectories is 1D rather than 3D. Nevertheless, we were able to obtain quite accurate numerical convergence even in the 3D case.

We also note that, despite the additional complexity of the trajectories in the spin-up-down case, the resulting dwell time for this interval window is quite close to that of the spin-up system. This can presumably be understood by the fact that both systems share identical probability distributions over time. The dwell times and their distributions are nevertheless different for the two spin systems—and it is, indeed, this very difference that earlier researchers [36] have proposed might be experimentally discernible.

## 5. Conclusions

The connection between quantum trajectories, and quantum time quantities of various kinds, has been explored previously [5,6,35,36,37,38]. However, in this paper, we extended the previous theory in various ways. Recent work [35,36] concentrating on spin-1/2 particle systems has focused on the *arrival time* quantity, which has been criticized from the perspective of its experimental validity [37,38]. The dwell time quantity may prove to be more reliable [2,7], in part because it derives from a *bona fide* Hermitian quantum operator $\hat{T}$, which, moreover, commutes with the Hamiltonian $\hat{H}$. Accordingly, other recent work by one of the present authors (Poirier) and coworkers [5,6] has concentrated on the relationship between quantum trajectories and dwell times—albeit only in the context of 1D time-independent stationary scattering applications, and for interval windows that extend across the entire scattering region. All of this previous work was based on unipolar quantum trajectories.

The present contribution generalizes the earlier work by extending the quantum-trajectory-based dwell time theory to multidimensional, time-dependent wavepacket applications, for particles with spin, and for arbitrary intervals. It is not only dwell times themselves that are developed, and computed for a benchmark application considered previously [36]; it is also quantum-trajectory-based dwell time *distributions* that are derived and computed—the latter being considered to be of significant experimental relevance [7,36]. Although at least two previous formulations exist for defining dwell time distributions—i.e., one based on the dwell time operator itself [7] and the other on the flux–flux correlation function [7,41]—these differ from each other, and also from the quantum trajectory-based distribution developed here. Thus, if dwell time distributions really do prove to have experimental relevance as has been suggested, it could be quite interesting to see what those experiments reveal—although we leave such speculation to other papers (and likely other authors).

Another way in which the present work differs from earlier contributions is that we consider—evidently for the first time—the connection between quantum dwell times and *bipolar* quantum trajectories. The seeds for this idea—and, indeed, many other ideas explored in the present work—were planted in our earlier 1D time-independent stationary scattering efforts [5,6]. There, it was discovered that unipolar quantum trajectory traversal times are not equal to quantum dwell times, but, rather, the two are related by a factor equal to the scattering transmission probability (Equation (Equation 18)). This suggests a more direct connection with bipolar quantum trajectories. Indeed, we have now discovered a relation between the bipolar trajectory traversal times and the average dwell time 〈τ〉—i.e., Equation (Equation 32)—that is so natural that we can take it as a *definition* of 〈τ〉.

Far from being a disappointing “second place” quantity, the average dwell time 〈τ〉 is arguably more important than τ proper. This assessment once again owes much to previous work [2,5,6,7], which relates not $\tau_{k}$ itself, but, rather, its nonoscillatory or average contribution $\langle \tau_{k}\rangle$, to other time-independent quantum time quantities (i.e., time delays and Smith lifetimes) that are directly linked to the scattering matrix. This earlier theory was developed for intervals across the entire scattering region, and is important because it includes both wave-based and trajectory-based determinations of 〈τ〉, for general scattering potentials. Whether, and how, these techniques may be applied to intervals extending only partway into the scattering region remains to be seen; however, the answers will prove highly important for future work. The reason is that all of the bipolar theory developed in this paper applies only to the special case of free-particle applications. Going forward, we will wish to apply Equation (Equation 32) “in reverse”—i.e., to compute $\tau_{\pm }$ from 〈τ〉. If the latter can indeed be reliably computed across an arbitrary interval within the scattering region, and if it is found to be nonoscillatory throughout, then we have a highly promising means of defining bipolar quantum trajectories for arbitrary scattering potentials—a longstanding goal of one of the authors [27,28,29,30,31,32].

In the meantime, both the unipolar and bipolar quantum trajectory treatments appear here to have “proven their worth” with respect to computing dwell time quantities, even for highly complex situations involving multidimensional wavepacket dynamics for particles with spin—albeit all of it, thus far, under the simplifying assumption of free-particle dynamics. In any event, the many results presented in Section 4 reveal interesting insights into the behavior of the various kinds of dwell times and quantum trajectories. In particular, bipolar quantum trajectories provide us with not just one, but *two* distinct new dwell time distributions, to be possibly thrown into the experimental mix. Of the two types of quantum trajectories, the bipolar trajectories may well prove superior to unipolar trajectories—not only because they avoid the oscillatory behavior of the latter (which, admittedly, is not an issue for the present applications), but also because they may provide robust dwell time values in situations where the latter cannot.

It is worth addressing this last point in a bit more detail. As discussed in Section 2.2.4, unipolar dwell times are guaranteed to be nondivergent *only* for wavepackets whose Fourier density approaches zero as k→0 [7,57]. For the present application, this condition was satisfied, but only by virtue of the fact that a first-excited harmonic oscillator state was used for $\psi_{0z}\left(z\right)$. What if a ground state were used instead—i.e., a Gaussian wavepacket? The unipolar dwell time τ could diverge in this case; unipolar quantum trajectories near z=0 approach infinite traversal times, and unlike the first-excited state case, this is not mitigated by vanishing probability density as z→0. At the very least, there will be computational difficulties, as only a tiny fraction of unipolar trajectories contribute to τ (as evidenced even in this work, e.g., for the [0, 0.4] interval window). Evidently, this is not a problem for bipolar waves and trajectories, which pass through even the z≈0 region with finite velocity, and, thus, should yield finite dwell times, $\tau_{\pm }$. In any event, more analysis is certainly needed here.

## Figures and Tables

**Figure 1 entropy-26-00336-f001:**
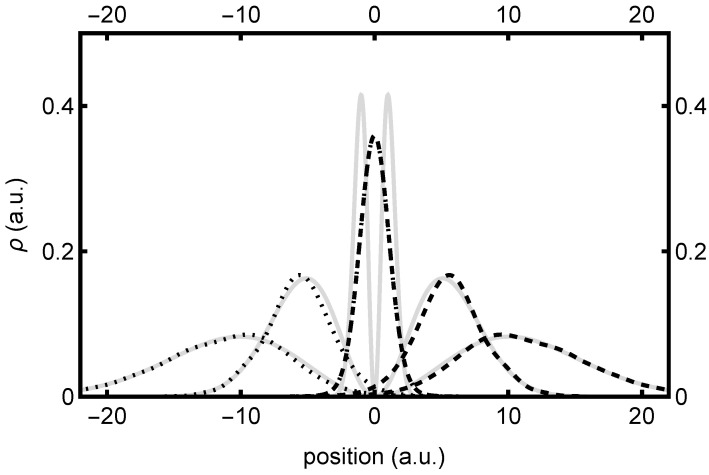
Plots of the unipolar probability density $|\psi_{z}{\left(z,t\right)|}^{2}$ (gray) and bipolar probability densities $|\psi_{+}{\left(z,t\right)|}^{2}$ (dashed) and $|\psi_{-}{\left(z,t\right)|}^{2}$ (dotted), at times t=0,5,10 versus position *z*, in arbitrary units (a.u.). For t=0, the magnitude of the $|\psi_{z}{\left(z,0\right)|}^{2}={\left|\psi_{0z}\left(z\right)\right|}^{2}$ plot is halved in order to be completely visible within the plot.

**Figure 2 entropy-26-00336-f002:**
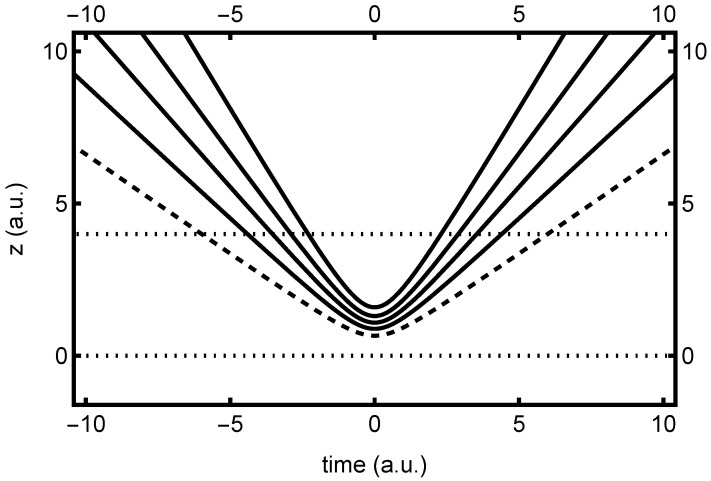
Plots of the unipolar quantum trajectories, z(C,t), associated with $\psi_{z}\left(z,t\right)$, versus time *t*, in arbitrary units, for various values of $C=\frac{1}{6}$ (dashed), $\frac{1}{3}$, $\frac{1}{2}$, $\frac{2}{3}$, and $\frac{5}{6}$. The window for the middle interval, [0, 4], is indicated with dotted horizontal lines.

**Figure 3 entropy-26-00336-f003:**
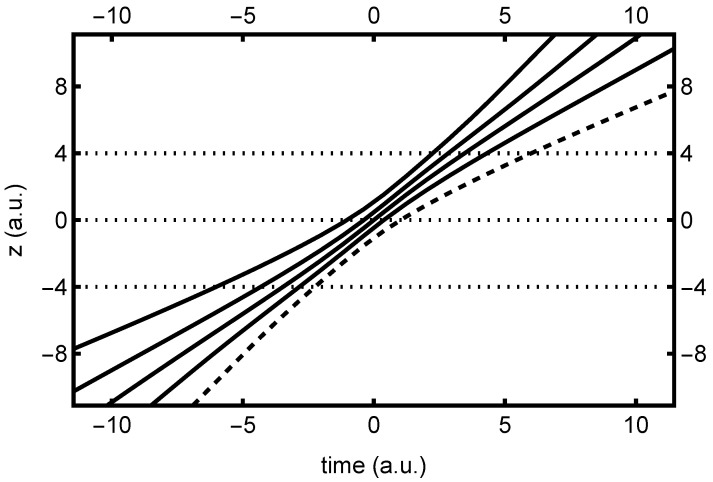
Plots of the bipolar quantum trajectories, $z_{+}\left(C_{+},t\right)$, associated with $\psi_{+}\left(z,t\right)$, versus time *t*, in arbitrary units, for various values of $C_{+}=\frac{1}{6}$ (dashed), $\frac{1}{3}$, $\frac{1}{2}$, $\frac{2}{3}$, and $\frac{5}{6}$. The window for the middle interval, [0, 4], is indicated with dotted horizontal lines—together with the “virtual” window at [−4, 0], used in the calculation of $\tau^{\pm }$ dwell times.

**Figure 4 entropy-26-00336-f004:**
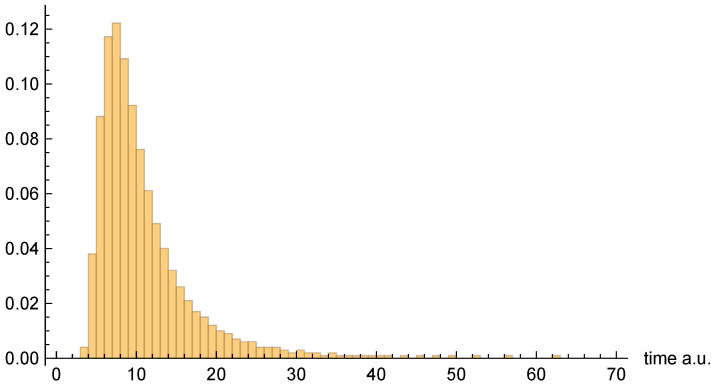
Dwell time distribution $\rho_{q}\left(2\tau \right)$ for interval [10, 20], computed using N=1000 unipolar quantum trajectories.

**Figure 5 entropy-26-00336-f005:**
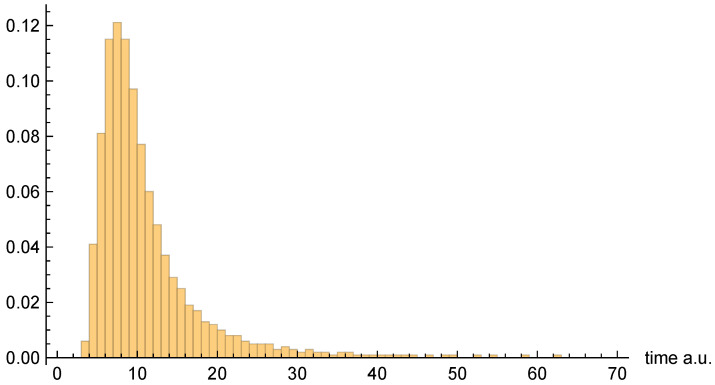
Dwell time distribution $\rho_{\pm }\left(\tau \right)$ for interval [10, 20], computed using N=1000 bipolar quantum trajectories.

**Figure 6 entropy-26-00336-f006:**
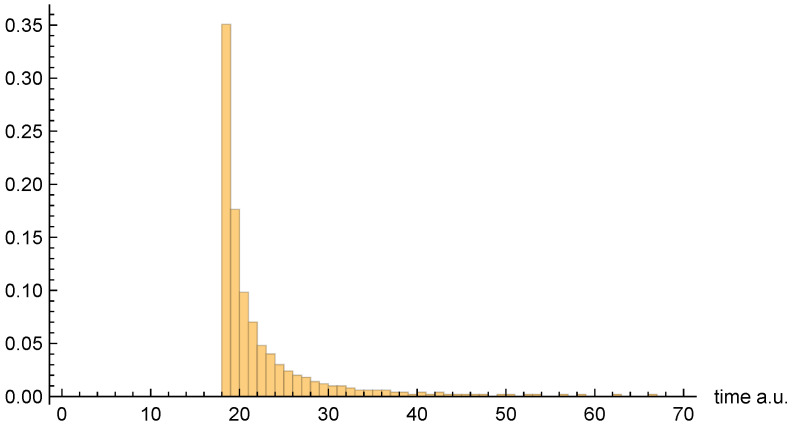
Dwell time distribution $\rho^{\pm }\left(\tau \right)$ for interval [10, 20] (plus virtual interval [−20, −10]), computed using N=1000 bipolar quantum trajectories.

**Figure 7 entropy-26-00336-f007:**
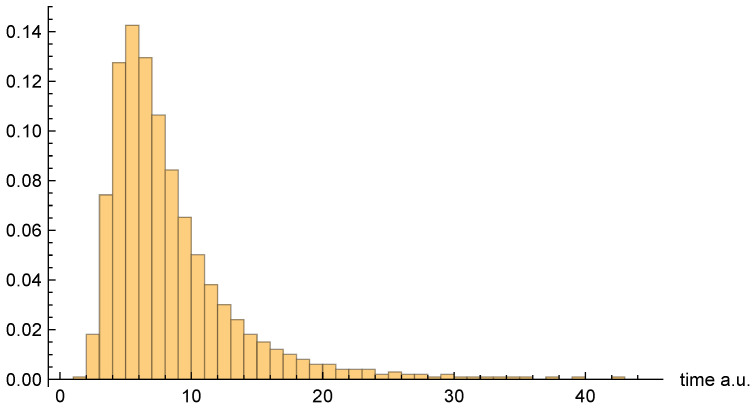
Dwell time distribution $\rho_{q}\left(\tau \right)$ for interval [0, 4], computed using N=1000 unipolar quantum trajectories.

**Figure 8 entropy-26-00336-f008:**
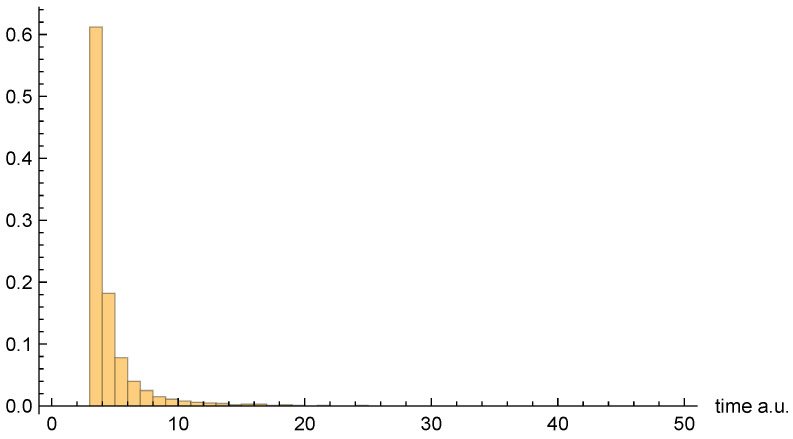
Dwell time distribution $\rho_{\pm }\left(\tau \right)$ for interval [0, 4], computed using N=1000 bipolar quantum trajectories.

**Figure 9 entropy-26-00336-f009:**
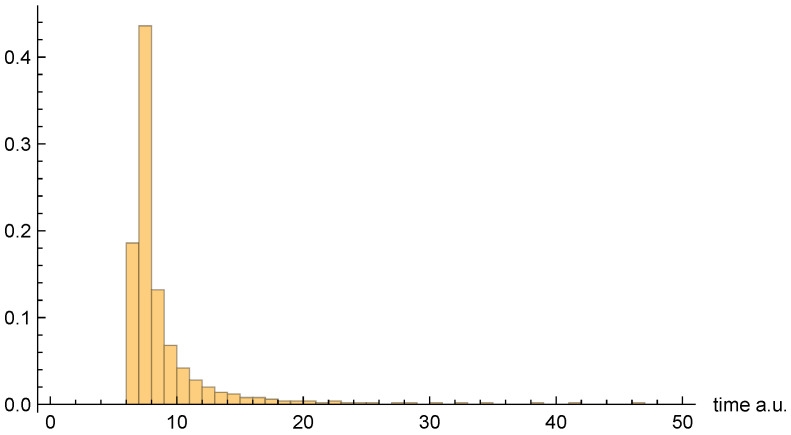
Dwell time distribution $\rho^{\pm }\left(\tau \right)$ for interval [0, 4] (plus virtual interval [−4, 0]), computed using N=1000 bipolar quantum trajectories.

**Figure 10 entropy-26-00336-f010:**
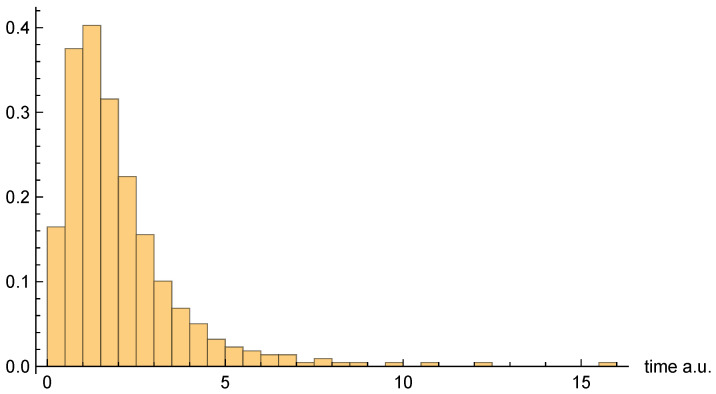
Dwell time distribution $\rho_{q}\left(\tau \right)$ for interval [0, 0.4], computed using just those 437 of *N* = 10,000 unipolar quantum trajectories that pass through the window.

**Figure 11 entropy-26-00336-f011:**
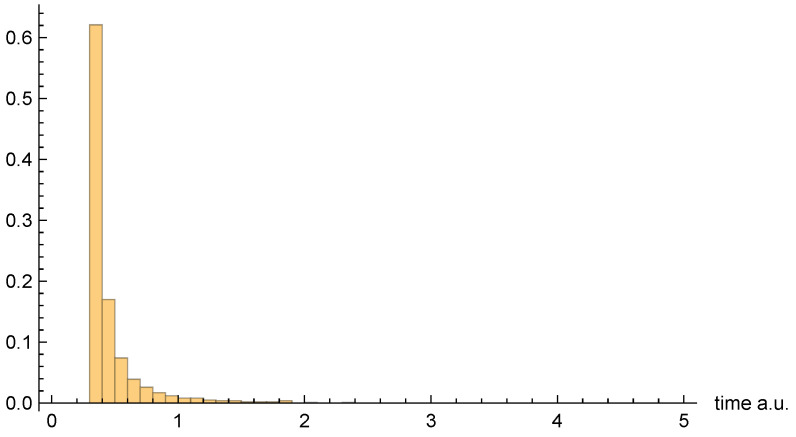
Dwell time distribution $\rho_{\pm }\left(\tau \right)$ for interval [0, 0.4], computed using N=1000 bipolar quantum trajectories.

**Figure 12 entropy-26-00336-f012:**
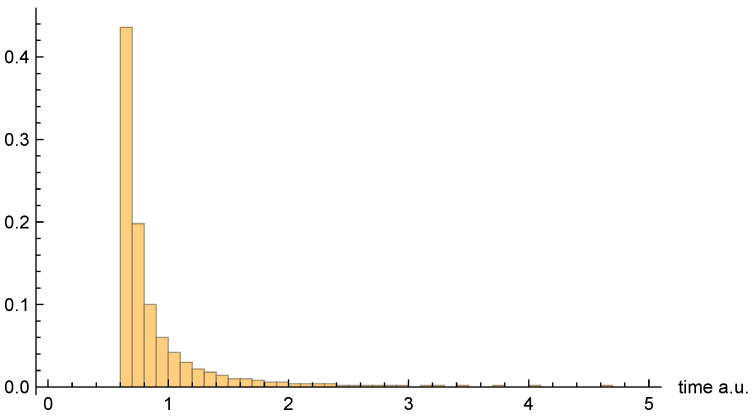
Dwell time distribution $\rho^{\pm }\left(\tau \right)$ for interval [0, 0.4] (plus virtual interval [−0.4, 0]), computed using N=1000 bipolar quantum trajectories.

**Figure 13 entropy-26-00336-f013:**
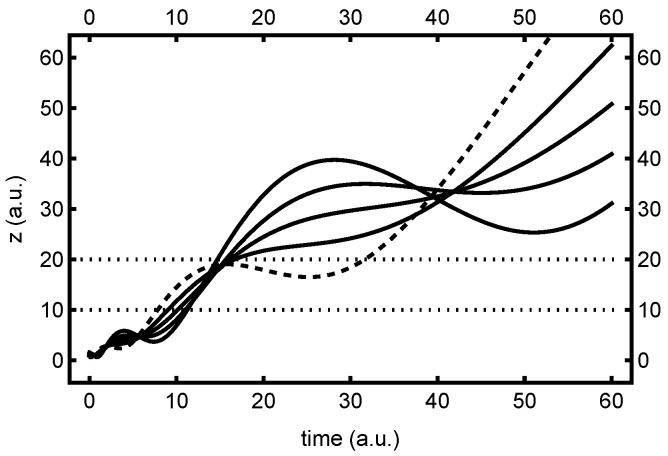
Plots of the *z* component of the spin-up-down unipolar quantum trajectories, z(C,t), versus time *t*, in arbitrary units, for various values of $C_{z}=\frac{1}{6}$ (dashed), $\frac{1}{3}$, $\frac{1}{2}$, $\frac{2}{3}$, and $\frac{5}{6}$. The window for the asymptotic interval, [10, 20], is indicated with dotted horizontal lines.

**Table 1 entropy-26-00336-t001:** Dwell times of various types (τ, $\langle \tau \rangle =\tau_{+}+\tau_{-}$, $\tau_{+}=\tau_{-}$, and $\tau^{\pm }$) and intervals $\left[z_{L},z_{R}\right]$, in arbitrary units, computed for the spin-up spin-1/2 system of Das and Dürr, using unipolar and bipolar quantum trajectory ensembles with N=1000 trajectories. The last two columns indicate standard deviations for two dwell time distributions: Δτ (for $\rho_{q}\left(\tau \right)$, column 6) and $\Delta \tau^{\pm }$ (for $\rho^{\pm }\left(\tau \right)$, column 7).

Interval	τ	$\langle \tau \rangle$	$\tau_{\pm }$	$\tau^{\pm }$	Δτ	$\Delta \tau^{\pm }$
[10, 20]	22.47	22.59	11.30	22.59	14.74	7.15
[0, 4]	8.672	9.034	4.517	9.059	6.000	4.438
[0, 0.4]	0.089	0.905	0.452	0.906	0.076	0.486

**Table 2 entropy-26-00336-t002:** Numerical convergence of dwell time τ for interval [10, 20], in arbitrary units, computed for the spin-1/2 system of Das and Dürr, using unipolar quantum trajectory ensembles with a range of trajectories, *N*. First two rows: spin-up case. Last two rows: spin-up-down case.

spin-up	*N* value	100	200	500	1000
case	τ value	21.99	22.20	22.38	22.47
spin-up-down	$N^{3}$ value	$5^{3}$	$20^{3}$	$46^{3}$	
case	τ value	19.82	21.58	21.40	

## Data Availability

All data generated by this project are available from the authors on request.

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
