# Peer review of "Dwell Times, Wavepacket Dynamics, and Quantum Trajectories for Particles with Spin 1/2"

_entropy, 2024, doi:10.3390/e26040336_

Round 1

Reviewer 1 Report

Comments and Suggestions for Authors

The paper generalizes the known theoretical connections between quantum trajectories and quantum dwell times for multi-dimensional time dependent wavepacket for spin 1/2 particles. The paper is generally well written, with sufficient background  provided and claims backed-up with well explored examples. While I am not an adherent to the idea of dwell times, I can see that the paper is an interesting contribution to the literature of quantum dwell times and to the problem of time in quantum mechanics in general. However, I cannot recommend its publication in its current form. The authors must make necessary revisions in accordance with the following comments:

  1. Page-2, lines 40-41. The authors mentions "'no-go'theorems'" that have been undone. There is no better example of such no-go theorem in the problem of time in quantum mechanics than Pauli's no-go theorem for the existence of self-adjoint time operators how it has been "undone" in (Galapon PRSA 458 2002). The authors are urged to reflect this fact.

  1. Page-2, lines-67-75. The author claims that "dwell time quantity is … more consistent than the arrival time.." and dwell time "offers more benefit"and "the dwell time operator has nicer properties". Dwell time and arrival time are two distinct aspects of time that they cannot be compared as to which one is better or offers more benefit than the other.  Each corresponds to a specific measurement, which the authors themselves discussed---the former to a "place to place" measurement and the later to a "time to place"measurement. The authors must revise the paragraph accordingly.

  1. Page-2, line 78.  It is better to use the acronym FFCF than ffcf. Ffcf looks like some random letters in print.

  1. The authors must provide an interpretation for equation-7.

  2. Page-4, lines 172-177. It is desirable to see the dwell time operator in its explicit form.

  3. It is not clear what is the relationship between equations 7 and 9. The same symbol is used in their left hand sides, but it is not apparent they are equal. The authors must clarify this.

  4. Page-5, line 203. What is <tau>? It is not defined prior this line. Is it the same <tau> introduced later in the paper?

  5. It is desirable to give references for equations 10 and 11.

  6. The last paragraph in the Conclusions is pompous and does not add anything of substance to the entire intentions of the paper. It can be deleted.
Comments on the Quality of English Language

Periods are missing in many places, especially after equations. 

Author Response

it is attached

Reviewer 2 Report

Comments and Suggestions for Authors

This manuscript represents an insightful, novel contribution to the study of dwell times within the context of the tunneling time problem. Overall I am pleased with the quality of the article and am happy to recommend it for publication in the journal Entropy. I have two very minor comments I wish to make on the contents of the article, which I believe will help to improve the quality of the final published version.

Firstly, in Equation 5, the notation |T(E)|2 is introduced to refer to the transmission probability (much later in the piece a similar notation is introduced for the reflection probability). Whilst this notation is standard in the study of tunneling times, here it seems out of place. The quantity T(E) on its own is never used or defined in its own right: |T(E)|2 is only defined with respect to ratios of fluxes. No other quantity is expressed in terms of its E-dependence, and its connection to k is not entirely straightforward. It should be made clearer that the transmission (reflection) probability used here is not defined in terms of transmission (reflection) amplitudes T (R), and perhaps alternate notation such as PT(k) for the probability of transmission should be considered.

Next, I think the authors would benefit from reading two articles by A. Steinberg and colleagues. The first, Phys. Rev. A 52, 32 (1995), discusses a conceptually distinct method by which one can define conditional dwell time distributions. The second, arXiv:2308.16069 (2023), experimentally shows the "interference between incident and reflected waves" inside a barrier that this manuscript discusses several times. 

These minor remarks aside, the manuscript is essentially fit for publication in its present form.

Author Response

it is attached
